# An archetype and scaling of developmental tissue dynamics across species

Yoshihiro Morishita [1,2,7] ✉, Sang-Woo Lee[1], Takayuki Suzuki[3],
Hitoshi Yokoyama [4], Yasuhiro Kamei [5], Koji Tamura[6] & Aiko Kawasumi-Kita[1,7]

Morphometric studies have revealed the existence of simple geometric relationships among various animal shapes. However, we have little knowledge of the mathematical principles behind the morphogenetic dynamics that form the organ/body shapes of different species. Here, we address this issue by focusing on limb morphogenesis in *Gallus gallus domesticus* (chicken) and *Xenopus laevis* (African clawed frog). To compare the deformation dynamics between tissues with different sizes/shapes as well as their developmental rates, we introduce a species-specific rescaled spatial coordinate and a common clock necessary for cross-species synchronization of developmental times. We find that tissue dynamics are well conserved across species under this spacetime coordinate system, at least from the early stages of development through the phase when basic digit patterning is established. For this developmental period, we also reveal that the tissue dynamics of both species are mapped with each other through a time-variant linear transformation in real physical space, from which hypotheses on a species-independent archetype of tissue dynamics and morphogenetic scaling are proposed.

A century ago, D'arcy Thompson explored how biological shapes and their diversity relate to the principles of physics and mathematics[1,2]. One of the most well-known aspects of his work is the degree to which differences in the shapes (or spatial arrangement of anatomical landmarks) of related animals could be described using relatively simple geometrical transformations; for instance, the shapes of two fish species were shown to be mapped onto each other by an affine/linear transformation (Fig. 1A). His study suggests the existence of a more or less standard shape across a group of species from which the distinct shape of each species is derived (Fig. 1A). Considering that many factors influence the determination of species-specific shapes, such as growth patterns, mechanical parameters, or boundary conditions, his findings - that the mapping of interspecies relationships is so

mathematically simple - were surprising. In addition, cases in which the mapping is linear have been studied in the context of scaling of the shapes of body parts across species. For example, among different species of Darwin's finch, beak shapes very closely align after applying a transformation represented by a diagonal or shear matrix, showing the interspecies scaling in the direction of each eigenvector[3,4].

While such beautiful interspecies geometric relationships are known for body shapes once formed[1,5], there is almost no quantitative information on the conservation and diversity of physical morphogenetic processes among species, i.e., the tissue deformation dynamics during animal development that include spatio-temporal patterns of area/volume change of each local tissue piece and the extent/direction of its stretching or shrinking[6]. During development of most organs, the

[1]Laboratory for Developmental Morphogeometry, RIKEN Center for Biosystems Dynamics Research, Kobe 650-0047, Japan. [2]Precursory Research for Embryonic Science and Technology (PRESTO) Program, Japan Science and Technology Agency, 4-1-8 Honcho, Kawaguchi, Saitama 332-0012, Japan. [3]Department of Biology, Graduate School of Science, Osaka Metropolitan University, Osaka 558-8585, Japan. [4]Department of Biochemistry and Molecular Biology, Faculty of Agriculture and Life Science, Hirosaki University, Aomori 036-8561, Japan. [5]Optics and Bioimaging Facility, Trans-Scale Biology Center, National Institute for Basic Biology, Myodaiji, Okazaki, Aichi 444-8585, Japan. [6]Department of Ecological Developmental Adaptability Life Sciences, Graduate School of Life Sciences, Tohoku University, Sendai 980-8578, Japan. [7]These authors contributed equally: Yoshihiro Morishita, Aiko Kawasumi-Kita. ✉e-mail: yoshihiro.morishita@riken.jp

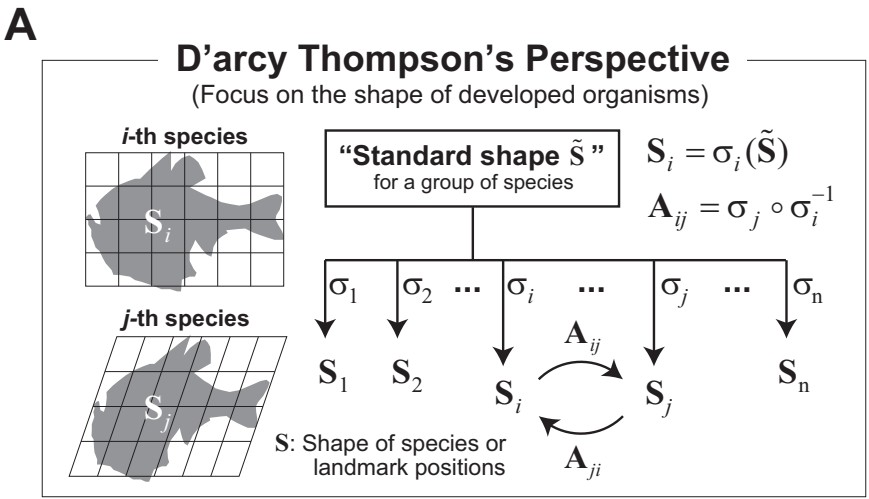

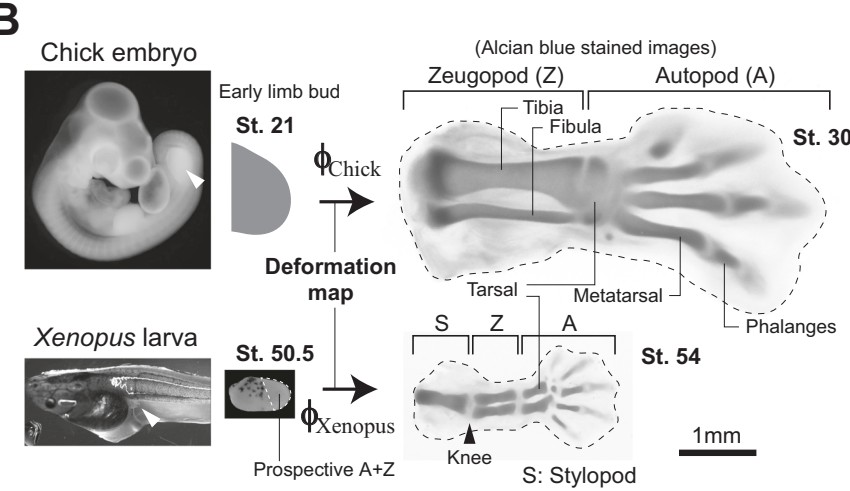

**Fig. 1 | The findings by D'arcy Thompson and their implications. A** D'arcy Thompson demonstrated that the shapes or spatial arrangement of anatomical landmarks of related animals can be mapped onto each other by relatively simple geometrical transformations (left panels). His study suggested the existence of a standard shape $\tilde{S}$ across a group of species from which the shape of each species $S_i$ is derived. Denoting it as $S_i = \sigma_i(\tilde{S})$, Thompson's work states that the interspecies transformation $A_{ij} = \sigma_j \circ \sigma_i^{-1}$ could be relatively simple. The fish silhouettes were traced from diagrams in Thompson's work[1]. **B** Hindlimb morphogenesis in chick and *Xenopus laevis*. The *Xenopus* limb bud is about half the size of that in chick for each axial direction (note that the scale bar is common for both species). Also, the absolute size, aspect ratio, and outline shape of each primordium differ among species. On the other hand, spatial patterns of gene expression and spatial arrangements of anatomical structures relative to the entire body/organ often share similarities among species, especially in earlier developmental phases[7,8]. This suggests the existence of organ-specific conserved dynamics of tissue growth and deformation across species (or a kind of archetype of dynamics), which might be observable after appropriate geometric transformations that cancel factors related to generating interspecies differences. In addition, there might be a developmental time window during which tissue dynamics are conserved and outside of which interspecies difference/diversity of the dynamics are pronounced.

anatomical regions formed from the initial limb buds differ between the two species. In *Xenopus*, all limb structures, i.e., the autopod (A), zeugopod (Z), and stylopod (S), are formed from the limb buds (the dotted region in a limb bud from St. 50.5 shows the prospective autopod and zeugopod regions). In chick, only the former two are formed, and the stylopod is embedded in the trunk. Skeletal patterning with cartilage differentiation becomes obvious around St. 54 for *Xenopus* and St. 30 for chick. The proportion of each skeletal segment (in particular, the tibia/fibula and tarsal) and the number of digits differ between these species. White arrow heads: hindlimb bud.

To address this issue, it is necessary to obtain quantitative tissue deformation maps (i.e., the positional correspondence of each point within a tissue at different time points or the trajectory of each point by which deformation characteristics can be calculated) for the developmental processes of organs of interest for each species, which makes it much more difficult than morphometric comparisons of shapes that have already reached their mature form, as typified by D'arcy Thompson's study. With current measurement techniques, in toto imaging at single-cell resolution is limited to small, thin tissues from very early stages in the development of embryos/organs that can be cultured under a two-photon or light-sheet microscope[9–12]. However, it has been demonstrated that tissue dynamics can be reconstructed by complementing the lack of resolution during deep tissue imaging of the different phases of organ development with statistical methods[13–16].

In this study, using vertebrate limb development as an example, we quantitatively compare the tissue dynamics between two species in different classes of animals, *Gallus gallus domesticus* (chicken) and *Xenopus laevis* (African clawed frog), to explore the geometric relationship between them (Fig. 1B). Both species share many developmental characteristics, including major signaling and gene expression patterns, whereas the trigger and timing of development are clearly different. Limb buds of amniotes including chick develop concurrently with the main body axis formation of an embryo and arise from the

lateral plate mesoderm. In contrast, limb development in *Xenopus* proceeds as one of the thyroxine (thyroid hormone)-dependent events in metamorphosis after embryonic stage[17], and the precise origin of a limb bud is difficult to determine[18]. Through the comparison of homologous organs with such qualitative differences, we inquired into the existence of archetypal tissue dynamics. We began by measuring the lineages of mesenchymal cells in the *Xenopus* limb to obtain its deformation map by applying the Bayesian method previously established using data from chick limb measurements[13,14]. We then propose the concepts of rescaled tissue dynamics and synchronization of developmental clocks, which enable direct comparisons of the morphogenetic dynamics of homologous organs with different sizes/shapes and developmental rates.

## Results

### Reconstruction of *Xenopus* hindlimb tissue deformation maps

To trace the lineage of mesenchymal cells in a developing limb bud, we used transgenic *Xenopus laevis* harboring the enhanced green fluorescent protein (EGFP) gene under the control of a heat shock promoter (Methods). Local irradiation using an infrared laser can locally increase tissue temperatures allowing for the induction of EGFP expression in cells contained within each irradiated spot, which on average spans just a few cells[19,20] (Fig. 2A, Methods). For each transgenic individual, we labeled 50 to 100 points on the frontal plane around the mid dorso-ventral (D-V) level in a gridded manner and measured their positions using two-photon microscopy (Fig. 2A, Supplementary Fig. 1, and Methods). Given that the expression of EGFP induced by a single heat shock stimulus is transient due to dilution by cell division and protein degradation, and the fluorescent signal is significantly weakened after several days in the case of tadpole mesenchymal cells, it is difficult to track the lineages of labeled spots in a single tadpole over the entire period of interest (i.e., from the initial budding of the limb to the stage at which skeletal patterning is complete). Therefore, we combined the data from 11 individuals at different developmental stages (the measurement intervals of some individuals overlapped) to cover the entire period of interest (Fig. 2B and Supplementary Fig. 1). The developmental stage (here termed $t_{Xenopus}$) of each individual at each measurement time point was determined morphometrically based on the outline of its shape, and each was resized according to the mean growth curve as performed in a previous study (Methods)[21]. Using these data, we reconstructed tissue deformation maps for nine consecutive time intervals using the previously proposed Bayesian method, and integrated them to obtain the full dynamics, $\mathbf{x} = \boldsymbol{\phi}_{Xenopus}(\mathbf{X}, t)$, where $\mathbf{x}$ is the positional vector at time $t$ of a cell initially located at $\mathbf{X}$ (Methods, Fig. 2C, Supplementary Fig. 1)[13].

From the reconstructed maps, we quantified the spatiotemporal patterns of local deformation, specifically the area growth rate and deformation anisotropy (direction-dependent tissue stretch/shrinkage), for nine consecutive time intervals during the period from $t_{Xenopus} = 50.6$ to 54.4 (Fig. 2D). For comparison, the deformation patterns of the chick hindlimb mesenchyme during the period from $t_{Chick} = 21$ to 30.5 that we previously reported were replotted with minor modifications in Fig. 2E (Methods)[13]. It should be noted that the (prospective) anatomical region contained in the initial limb bud is different between *Xenopus* and chick (Fig. 1B); *Xenopus* limb buds include the prospective autopod (toe-to-ankle), zeugopod (lower leg), and stylopod (upper leg) regions[22]. In contrast, as shown later, based on the inverse mapping of cartilage patterns, chick limb buds contain mainly the former two regions, while the stylopod is embedded in the trunk. Thus, in the following analyses, we mainly compared tissue dynamics for the prospective autopod and zeugopod regions between the two species.

Importantly, two characteristic tissue dynamics identified in previous analyses of chick limb development were also found in *Xenopus*. One is antero-posterior (A-P) asymmetric growth, especially prominent in autopod formation. This is evident in the heatmaps of tissue growth rates during $t_{Xenopus} = 52–53$ and $t_{Chick} = 23-25$ (Fig. 2D, E [top]), and regions with higher growth rates (reddish regions) correspond well with the expression pattern of SHH, which is involved in the regulation of cell proliferation[23]. In fact, the posterior half of the autopod in a more mature limb bud with a clear skeletal pattern ($t_{Xenopus} = 54.4$ or $t_{Chick} = 30.5$) is derived from the smaller posterior portion of the early limb bud ($t_{Xenopus} = 50.6$ or $t_{Chick} = 21$; Fig. 2F and Supplementary Fig. 2). Further, in our previous study on chick limb development, we showed that this A-P asymmetric area growth rate at the tissue level is quantitatively consistent with the positional dependence of the cell cycle time[14]. The other characteristic is that differential or distally biased growth is not the primary mechanism of limb bud elongation[14,24]. Regions with high area growth rates are not necessarily confined to the distal end of the limb bud (Fig. 2D, E [top]). Instead, deformation anisotropy is high and proximo-distally (P-D) oriented in a wide span of the limb tissue (Fig. 2D, E [bottom]), meaning that each local tissue piece or cell subpopulation changes its shape as it extends along the P-D axis at a similar (i.e., spatially homogeneous) rate, which is further supported by the analysis detailed in the next section. This fact indicates that, similar to the chick case[14], the P-D elongation of a *Xenopus* limb bud cannot be explained by the classical model that limb bud elongation is caused primarily by proliferation of distal cells[25–27] (see also Boehm et al. (2010) that nicely reviews the history of proliferation gradient model); it should be noted that the factors that drive this anisotropic local tissue deformation remain unknown. Taken together, these results demonstrate that the tissue dynamics during limb morphogenesis are similar between chick and *Xenopus*.

### Decomposition into average growth and rescaled dynamics

To discuss similarities/differences in dynamics between species in more detail, we need to develop an analytical method for direct and quantitative comparison of the deformation dynamics of tissues of different sizes and perimeter shapes. Here, we attempted to address this issue by changing the coordinate system under which the dynamics are observed. The expression patterns of key genes common to homologous organ development across species may provide clues to selection of such coordinates. For example, in each anatomical region, corresponding marker genes including homeobox genes are expressed to designate a specific address within a tissue (e.g., *Hoxa13* for autopods and *Hoxa11* for zeugopods[28]). Considering that the size of their expression domains is not absolute across species, but relative to or scale with overall tissue size, it suggests that the essence of the morphogenetic process may be captured by re-observing tissue dynamics based on the change in relative position of each cell.

The situation is simple in the case of one-dimensional (1D) tissue deformation: the position relative to the whole tissue (denoted by ξ) can be uniquely defined as the position rescaled by the tissue size or length at each timepoint $t$, $l(t)$, after subtracting the coordinate for the tissue's geometric center (Fig. 3A). In the ξ-space, tissue deformation is expressed as a velocity field that represents the change in rescaled cell position (black arrows in Fig. 3A) and the velocity field reflects the spatial pattern or heterogeneity of the tissue growth rate. Under spatially uniform growth, as a special case, the rescaled position of each cell is time-invariant, i.e., no flow appears. Thus, 1D growth patterns between tissues/species with different overall sizes can be compared by measuring the degree of match between velocity fields within the ξ-space.

In contrast, in multi-dimensional (2D or 3D) cases, since the shape as well as the size of a tissue change with its deformation, its rescaled position is not uniquely defined. Considering that 1D tissue size $l(t)$ is the product of the initial size and the spatial average of local growth, the multi-dimensional position rescaled by the product of the initial tissue size (e.g., centroid size $C_0$) and the spatial average of the

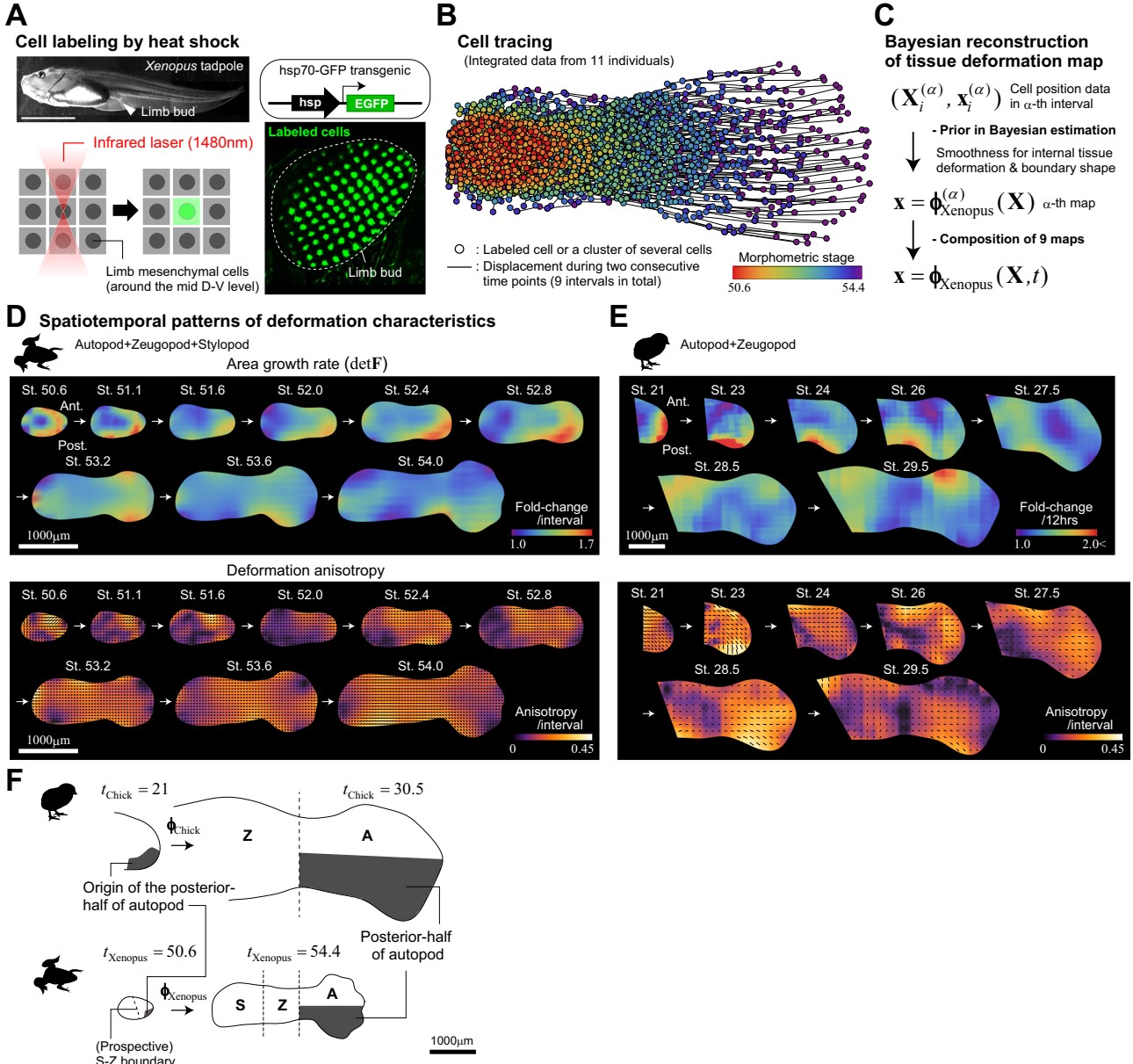

**Fig. 2 | Reconstruction of the tissue deformation map for *Xenopus* hindlimb development. A** Cell labeling by heat shock induction of EGFP expression. Mesenchymal cells around the frontal plane of mid D-V level were irradiated with an infrared laser using the IR-LEGO system. **B** Integrated data for mesenchymal cell lineages obtained from 11 individuals after morphometric staging and resizing based on mean growth curves. **C** The workflow for reconstructing the *Xenopus* limb deformation map (see Methods for details). **D**, **E** Spatiotemporal patterns of local tissue deformation for *Xenopus* (**D**) and chick (**E**). The patterns of area growth (top) and deformation anisotropy (bottom) are plotted; the chick pattern is a replot of our previous report with minor modifications[14]. In the bottom panels, the black line segments indicate the orientation of anisotropy (the length of each segment reflects the magnitude of anisotropy). **F** The posterior half of the autopod in a later limb bud with a clear skeletal pattern ($t_{\mathrm{Xenopus}} = 54.4$ or $t_{\mathrm{Chick}} = 30.5$) is derived from the smaller posterior portion of the early limb bud ($t_{\mathrm{Xenopus}} = 50.6$ or $t_{\mathrm{Chick}} = 21$), clearly showing A-P asymmetric growth; top: chick, bottom: *Xenopus*. Note that the images of the *Xenopus* limb bud, except for the top-left photo in panel (**A**), are inverted in the A-P direction to have the posterior side facing downward. Source data are provided as a Source Data file.

deformation gradient tensor representing local tissue growth (denoted by $\bar{\mathbf{F}}(t)$) through its inverse operation (i.e., the position relative to the whole tissue after canceling out average tissue growth) is a natural extension of the rescaled position in the 1D case (Fig. 3B; see also Methods for mathematical details):

$$\boldsymbol{\xi}(\boldsymbol{\xi}_0(\mathbf{X}), t) = \mathbf{L}^{-1}(t)(\mathbf{x}(\mathbf{X}, t) - \bar{\mathbf{x}}(t)), \qquad (1)$$

where $\boldsymbol{\xi}_0 \equiv \mathbf{X}/C_0$ is the $\boldsymbol{\xi}$-representation of the initial position $\mathbf{X}$, and $\mathbf{L}(t) \equiv C_0\bar{\mathbf{F}}(t)$ is interpreted as multi-dimensional tissue size. $\bar{\mathbf{x}}(t)$ is the position of the geometric center of the tissue at $t$ under the ordinary

Cartesian coordinate system. Under biologically plausible assumptions, the velocity field in the $\boldsymbol{\xi}$-space that represents the rescaled tissue deformation dynamics obeys the following equation:

$$\frac{\partial \boldsymbol{\xi}}{\partial t}(\boldsymbol{\xi}_0, t) = \mathbf{L}^{-1}(t)\left(\bar{\mathbf{S}}_{\mathbf{X}}(t) - \bar{\mathbf{S}}(t)\right)\mathbf{L}(t)\boldsymbol{\xi}(\boldsymbol{\xi}_0, t) + \mathbf{v}_g(\boldsymbol{\xi}_0, t). \qquad (2)$$

$\bar{\mathbf{S}}_{\mathbf{X}}(t)$ and $\bar{\mathbf{S}}(t)$ in the first term on the right side represent a local average of the velocity gradient tensor associated with the initial position of a focal cell $\mathbf{X}$ and the global average of the velocity gradient across the whole tissue, respectively. This first term clearly shows that

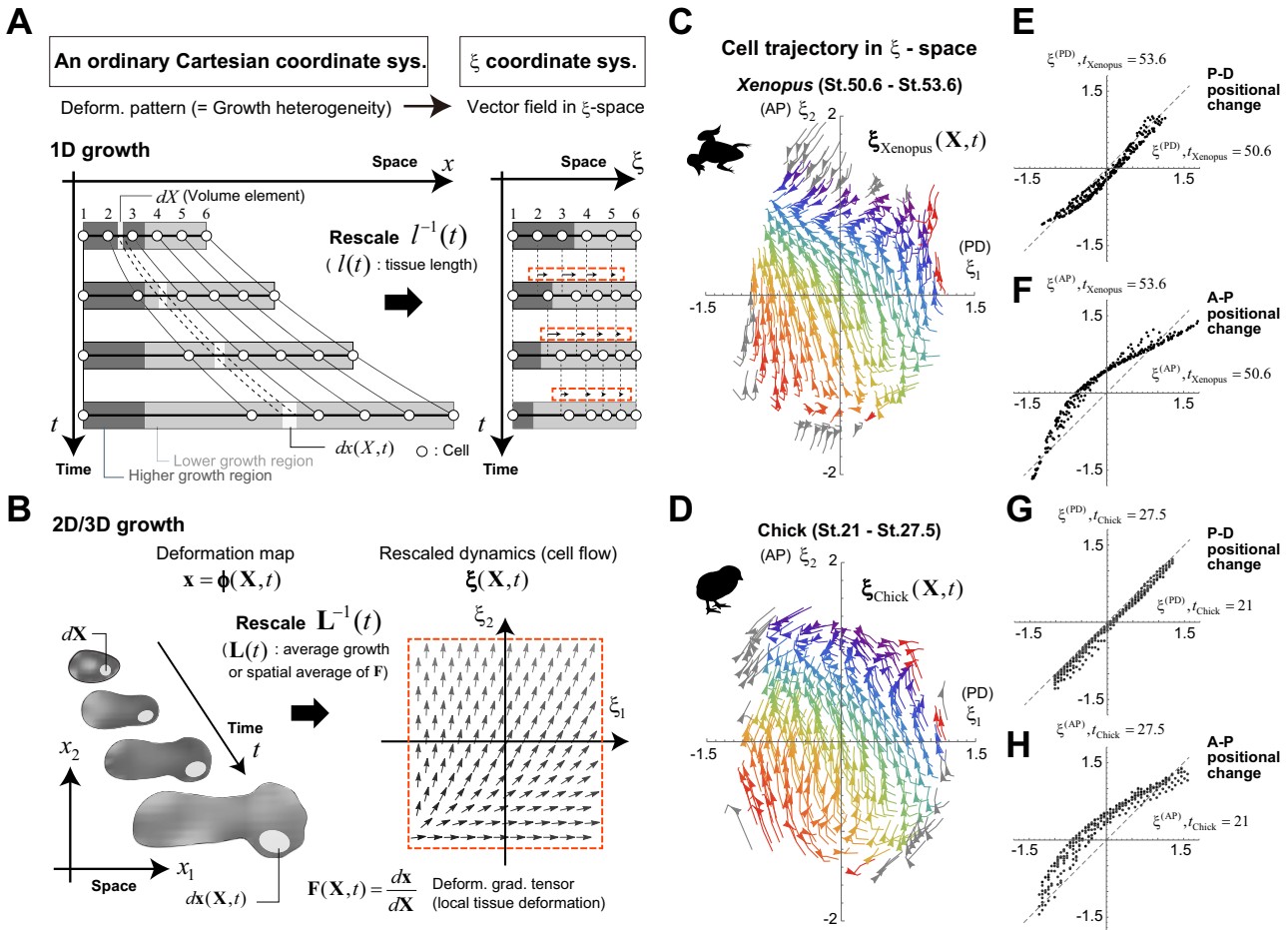

**Fig. 3 | Decomposition of tissue dynamics into average growth and rescaled dynamics. A, B** Tissue dynamics under a fixed Cartesian coordinate system for real physical space and under the **ξ**-coordinate system. **ξ**-coordinates are defined as the spatial coordinates rescaled by average growth, i.e., the spatial average of the volume growth rate or tissue length in a one-dimensional case (**A**) and that of the deformation gradient tensor in a multi-dimensional case (**B**). In the **ξ**-space, tissue dynamics are represented as cell flow reflecting the spatial heterogeneity of local deformation (see the red dotted boxes). **C–H** Cell trajectories in the **ξ**-space for *Xenopus* (**C**) and chick (**D**) limb morphogenesis and the positional changes in the proximo-distal (P-D) (**E** *Xenopus*; **G**: chick) or antero-posterior (A-P) (**F**: *Xenopus*; **H**:

chick) direction in the **ξ**-space between an early stage and a stage in which digit patterning is fairly well established (see also Fig. 4C). Arrowheads show the orientations of the flows at $t_{Xenopus} = 52.4$ for *Xenopus* and $t_{chick} = 24$ for chick. The colored regions indicate the initially overlapping regions of the limb buds from both species in the **ξ**-space. For both species, trajectories of the same color indicate trajectories from the same initial position. The cell flows in the **ξ** coordinate system shown in panels (**C**) and (**D**) correspond to the morphogenesis of the region consisting of prospective autopods and zeugopods. Source data are provided as a Source Data file.

the rescaled position of the focal cell changes depending on the difference between these two factors, that is, the spatial heterogeneity in local tissue deformation. The second term in the equation describes the effect of the dynamics of the tissue geometric center (Methods). When $\bar{\mathbf{S}}_{\mathbf{X}}(t) = \bar{\mathbf{S}}(t)$ holds at any location $\mathbf{X}$, i.e., spatially uniform growth, no flow appears.

Using the ξ representation defined above, the tissue deformation maps for chick and *Xenopus* limb morphogenesis reconstructed in the previous section were decomposed into their average growth and rescaled dynamics as follows:

$$\boldsymbol{\phi}_{Chick}(\mathbf{X},t) = \mathbf{L}_{Chick}(t)\boldsymbol{\xi}_{Chick}(\boldsymbol{\xi}_0,t) + \bar{\mathbf{x}}_{Chick}(t)$$
$$\boldsymbol{\phi}_{Xenopus}(\mathbf{X},t) = \mathbf{L}_{Xenopus}(t)\boldsymbol{\xi}_{Xenopus}(\boldsymbol{\xi}_0,t) + \bar{\mathbf{x}}_{Xenopus}(t) \tag{3}$$

The spatiotemporal patterns of cell trajectories in the **ξ**-space for both species showed high similarity especially during the stages prior to full differentiation of digit cartilage within the digit-forming region (Fig. 3C, D for the temporal windows of $t_{Xenopus} \in [50.6, 53.6]$ and $t_{Chick} \in [21, 27.5]$; see also Supplementary Fig. 3 for the full time periods and Supplementary Software 1); (i) the motion in the P-D direction was small, representing nearly uniform growth along the P-D axis, while (ii)

the flow from posterior to anterior was pronounced, showing clear A-P asymmetric growth and deformation in both species. Furthermore, plotting the correspondences of the P-D or A-P coordinates in the **ξ**-space between early and later stages for each species also quantitatively supports these characteristics (Fig. 3E–H). Although more direct comparisons will only be possible after synchronizing the developmental clocks between species as shown below, we can intuitively see that the cell trajectories (the flow patterns in the **ξ** coordinate system) of the internal tissues that will form future skeletal structures are more consistent between species (i.e., basically oriented from posterior to anterior) than those near the tissue boundaries (Fig. 3C, D); the latter likely contributes to the generation of differences in the contour shapes of the limb buds. Thus, introducing ξ-coordinates enabled comparison of the tissue dynamics of homologous organs between different species on the same scale.

### Cross-species synchronization of developmental clocks
To enable full comparison of tissue dynamics in the **ξ**-space, we then attempted to synchronize the developmental clocks between the two species. The degree of developmental progression of each species is generally represented by stages, which are determined based on

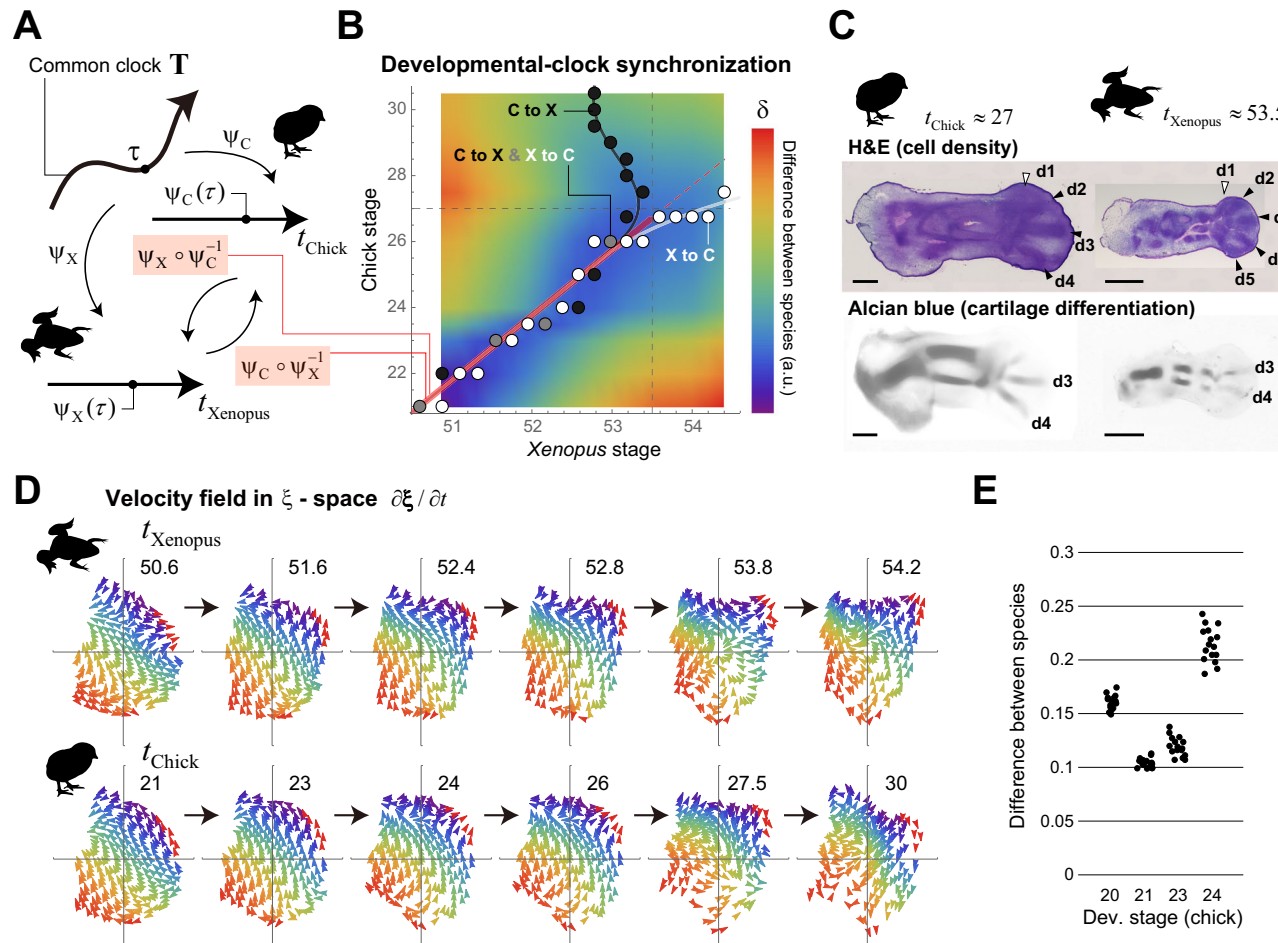

**Fig. 4 | Cross-species synchronization of developmental clocks. A** Definition of the common clock and relationship between developmental stages of different animals (see Methods for details). **B** Synchronization of the developmental clocks between *Xenopus* and chick. The color of the heatmap shows the distance of each pair of developmental stages (see Methods for details). The white and gray points/curve indicate the correspondence of each *Xenopus* stage to the closest chick stage, which defines the map $\psi_C \circ \psi_X^{-1}$. The black and gray points/curve shows the map $\psi_X \circ \psi_C^{-1}$. Around the ranges $t_{Xenopus} \in [50.6, 53.5]$ and $t_{Chick} \in [21, 27]$, both stages correspond linearly and in a one-to-one manner (red segment). **C** (top) In terms of the spatial heterogeneity in cell density (Hematoxylin-Eosin (H&E) staining), digit patterning is well established by approximately $t_{Chick} = 27$ for chick and $t_{Xenopus} = 53.5$ for *Xenopus*. (bottom) The timing of cartilage differentiation (Alcian blue staining) is slightly delayed compared to the patterning of cell density. The black arrowheads indicate the already formed digit regions, while the white ones show the digit regions just beginning to form or that have yet to begin formation. d:

digit. Scale bar: 500 μm. **D** Temporal changes in the cell velocity fields for both species. Until the stage at which anatomical patterning is well established, the orientation pattern of cell flow in the **ξ** space, especially in the internal tissues, is very similar between the species. In contrast, at later stages, differences in the orientation pattern become more pronounced (see also Supplementary Fig. 3). Each pair of $(t_{Xenopus}, t_{Chick}) = (50.6, 21), (51.6, 23), (52.4, 24), (52.8, 26)$ represents almost the same timepoint of the common clock. **E** Dependence of the agreement in cell trajectories in the **ξ**-space on the selection of initial stages; we tested the initial time point for *Xenopus*, which was fixed at $t_{Xenopus} = 50.6$, and the four initial time points for chick which were $t_{Chick} = 20, 21, 23, 24$. (Median, SD) = (0.160, 0.0067), (0.103, 0.0044), (0.118, 0.0090), (0.213, 0.0162) for $t_{Chick} = 20, 21, 23, 24$, respectively. See Methods for details including information on SD and Supplementary Fig. 4 for the case in which the initial time point for chick was fixed. Source data are provided as a Source Data file.

changes in appearance and characteristic events, e.g., Hamburger-Hamilton staging for chick[29] and Nieuwkoop-Faber staging for *Xenopus laevis* are typically used[30]. Such traditional species-specific staging values/scales do not necessarily correspond across species. Thus, to synchronize developmental time, we introduce a new concept of a common clock **T**, which is defined as a directed curve or one-dimensional manifold, and each point τ on **T** corresponds to a certain timepoint on the common clock (Fig. 4A and Methods). **T** is an abstract concept with no concrete scale, and here we interpret a set of traditional developmental staging values for a species as a time coordinate system ($t_{Species}$) for this common clock (to be precise, the original staging scale uses only integer values, but we considered an extension to use real values). Mathematically, this corresponds to species-specific mapping $\Psi_{Species}(\tau)$ from **T** to one-dimensional Euclid space **R** (Fig. 4A). Inversely, a point on **T** or a common time point

corresponding to a specific stage $t_{Species}$ is designated as $\tau = \psi_{Species}^{-1}(t_{Species})$. Under this setting, synchronizing the developmental clocks between two species (α and β) is equivalent to determining the correspondence of their stages with each other via the common clock **T**, i.e., the identification of $t_\alpha = \psi_\alpha(\tau)$ and $t_\beta = \psi_\beta(\tau)$ for each common timepoint τ and the definition of a composite map (or a transformation of time coordinate for **T**) $t_\beta = \psi_\beta \circ \psi_\alpha^{-1}(t_\alpha)$ or $t_\alpha = \psi_\alpha \circ \psi_\beta^{-1}(t_\beta)$ (Fig. 4A).

As a way to find the composite map that relates stages between chick and *Xenopus* (denoted by $\psi_C \circ \psi_X^{-1}$ and $\psi_X \circ \psi_C^{-1}$), we aligned the scales of their temporal axes such that the mean difference in cell trajectories from the same initial position in the ξ-space is minimized (see Methods for details). Over a wide span of time, for each developmental stage in one species, there is a corresponding stage in the other species at which the positions along the cell trajectories are

highly consistent (blue region in the heatmap in Fig. 4B); the white (and gray) points give $\psi_C \circ \psi_X^{-1}$ and the black (and gray) points give $\psi_X \circ \psi_C^{-1}$ in Fig. 4B. Assuming that the common clock **T** runs smoothly, the composite maps should be bijective and continuous. In the case of chick and *Xenopus*, those conditions are satisfied up to around $t_{Xenopus} = 53.5$ and around $t_{Chick} = 27$, by which point digit patterning is fairly well established in terms of spatial heterogeneity in cell density, although the timing of cartilage differentiation is slightly delayed compared with cell density patterning (Fig. 4C). In particular, within the ranges $t_{Xenopus} \in [50.6, 53.5]$ and $t_{Chick} \in [21, 27]$, both stages correspond linearly (red line in Fig. 4B). Further, the blue area of the heatmap narrows (like an hourglass) at around chick stage 23-24 and *Xenopus* stage 52-52.5. In those stages, A-P asymmetric tissue growth clearly appears, and it greatly affects the change in the rescaled cell position in the ξ space (i.e., cell flow), meaning that those time windows should correspond between the two species.

In contrast, at later stages, the stage correspondences from chick to *Xenopus* and from *Xenopus* to chick do not match, probably because species-specific tissue deformation is beginning to appear by this time. For example, regarding cell/tissue behavior during digit formation within the paddle-like autopod region, amphibians and amniotes have been reported to have qualitative differences in the rate of cell death in the interdigital zone and in the segmentation processes of digits[31]. Therefore, using our approach, it is not possible to synchronize the developmental clock after anatomical patterning (in terms of the spatial heterogeneity in cell density) is established. It should be noted that, if viewed positively, using our tissue dynamics-based synchronization approach conversely could potentially enable the detection of the emergence of interspecies differences in tissue dynamics. This transition of similarity in tissue dynamics is also visually evident by comparing the velocity fields of cell flow in the ξ-space ($\partial \boldsymbol{\xi}/\partial t$) after synchronization of the developmental clocks (Fig. 4D).

Finally, it should be noted that the level of agreement in cell trajectories between species also depends on the choice of initial times. Based on the criterion to minimize the mean difference, the combination of chick stage 21 and *Xenopus* stage 50.6 was chosen, i.e., $\psi_X^{-1}(50.6) \approx \psi_C^{-1}(21)$ (Fig. 4E, Supplementary Fig. 4, and Methods).

### Conservation of rescaled tissue dynamics in τ-ξ spacetime

Under the synchronized developmental clocks, cell trajectories in the ξ-space show strong agreement between the two species (see the black dots in Fig. 5A for the comparison at $\tau = \psi_X^{-1}(53.6) \approx \psi_C^{-1}(27.5)$). Comparisons of the variability in tissue dynamics among individuals of the same species would be a good means of confirming this interspecies similarity. At present, however, it is quite difficult to perform long-term live imaging of cell trajectories in developing tissues and to quantify deformation maps for individual embryos. Instead, we used bootstrapped deformation maps as a measure for evaluating intraspecies variations. We generated 150 bootstrapped datasets from the original data for cell positional change that were used to calculate the chick hindlimb deformation map, and performed Bayesian reconstruction of the deformation maps for each bootstrapped dataset (denoted by $\boldsymbol{\phi}_{\mathrm{Chick}}^{\mathrm{BS},i}$, $i = 1, \cdots, 150$). Then, the rescaled dynamics were calculated for each bootstrapped map to assess the variability in cell trajectories within the same species (gray dots in Fig. 5A). Consequently, the interspecies differences (black dots in Fig. 5A) are comparable with the range of variation in the bootstrapped maps, strongly supporting the conservation of rescaled tissue dynamics between species.

For a visual understanding of this interspecies conservation, we generated a pseudo chick limb deformation map (denoted by $\tilde{\boldsymbol{\phi}}_{\mathrm{Chick}}(\mathbf{X}, \tau_1, \tau_2)$) by swapping the chick rescaled dynamics $\boldsymbol{\xi}_{\mathrm{Chick}}(\boldsymbol{\xi}_0, \tau)$ for the *Xenopus* dynamics $\boldsymbol{\xi}_{\mathrm{Xenopus}}(\boldsymbol{\xi}_0, \tau)$ as:

$$\tilde{\boldsymbol{\phi}}_{\mathrm{Chick}}(\mathbf{X}, \tau_1, \tau_2) = \mathbf{L}_{\mathrm{Chick}}(\tau_1)\boldsymbol{\xi}_{\mathrm{Xenopus}}(\boldsymbol{\xi}_0, \tau_2) + \bar{\mathbf{x}}_{\mathrm{Chick}}(\tau_1), \qquad (4)$$

where, for generality, different time variables are used for both species on the right side of the equation (i.e., $\tau_1$ for chick and $\tau_2$ for *Xenopus*); when both developmental clocks are in synchronization, $\tau_1 = \tau_2$ holds. The pseudo map was then compared with the true map $\boldsymbol{\phi}_{\mathrm{Chick}}(\mathbf{X}, \tau)$. To more easily understand the consistency of these maps, the prospective skeletal regions of the chick hindlimb at $t_{\mathrm{Chick}} = 21$ were calculated by inverse mapping from the already formed cartilage pattern at $t_{\mathrm{Chick}} = 30.5$ (Supplementary Fig. 5 and Methods), and the prospective regions were mapped by $\boldsymbol{\phi}_{\mathrm{Chick}}(\mathbf{X}, \tau)$ or $\tilde{\boldsymbol{\phi}}_{\mathrm{Chick}}(\mathbf{X}, \tau_1, \tau_2)$. Both of the resultant (partially presumptive) skeletal patterns at $\tau = \psi_C^{-1}(27.5)$ showed high consistency with each other (Fig. 5B [top], Supplementary Fig. 6 and Methods). Anatomically, the interspecies differences in the relative lengths of the tarsals and zeugopods and the number of formed digits begin to appear around $t_{\mathrm{Xenopus}} = 53.5$ (*xenopus*) and $t_{\mathrm{Chick}} = 27$ (chick). Thus, these highly consistent cell trajectories in the ξ-space suggest that temporal changes in the relative positions of cells within the developing tissues are well conserved regardless of the state of cartilage differentiation. Again, note that the synchronization of developmental clocks is essential; in swapping the rescaled tissue dynamics, the mismatch in the mapped skeletal patterns was significant when the developmental clocks were out of synchronization, i.e., $\tau_1 \neq \tau_2$ (Fig. 5B [bottom] and Supplementary Fig. 6).

Finally, to ensure that the conservation of the rescaled dynamics shown above only holds for anatomically corresponding regions, we compared the rescaled dynamics within different regions of the limb buds between species (Fig. 5C and Methods). As mentioned earlier, the early limb bud of the chick consists of the prospective autopod and zeugopod, whereas the *Xenopus* limb bud contains the additional prospective stylopod. We characterized the region within the early *Xenopus* limb bud using three parameters ($\Omega_X$ in Fig. 5C [left]), and measured the difference in rescaled dynamics in the overlapping regions of $\Omega_X$ and early chicken limb buds ($\Omega_C$). Two of the three parameters specified the orientation/axis of the tissue and the other parameter, the most important one, determined the spatial range along the P-D axis (Fig. 5C [left]). As the region $\Omega_X$ was restricted to the distal side, the mean difference in cell trajectories from the chick decreased, but the degree of overfitting showed a simultaneous sharp increase. Only in the parameter range that corresponds to the region $\Omega_X$ that consists of the autopod and zeugopod, a sufficiently small mean difference and overfitting are both achieved (green area in Fig. 5C [right]), clearly demonstrating that the consistency of the rescaled dynamics between species holds only for anatomically corresponding regions.

### Conservation of expression patterns for typical marker genes

It is by no means obvious that the spatiotemporal patterns of gene expressions coincide across species under the τ-ξ coordinate system, similar to the rescaled tissue dynamics as shown above, because that coordinate system is defined based only on tissue deformation maps, i.e., geometric (not molecular) information. To observe this, we performed RNAscope assays to visualize and compare marker genes for limb development, specifically, *Hoxa13*, *Hoxa11*, and *Hoxd13*, between chick and *Xenopus* (Fig. 6A, Supplementary Fig. 7 and Methods). The detected fluorescent signals were transformed into their ξ-representations at three different time points on the common clock (Fig. 6B). The expression ranges of these marker genes in the ξ-space and their temporal transitions basically showed a high degree of similarity between the two species except near the tissue boundaries. It should be noted that the overlap of *Hoxa11* expression ranges of both species was somewhat ambiguous in the early stages, probably because the stages correspond to the transient period in which the distal end of the *Hoxa11* expression region shifts from the tip of the limb bud to the more proximal side. Since *Hoxa13* and *Hoxa11* define the position of autopod and zeugopod formation, respectively, the similarity in gene expression patterns suggests that the encoding or

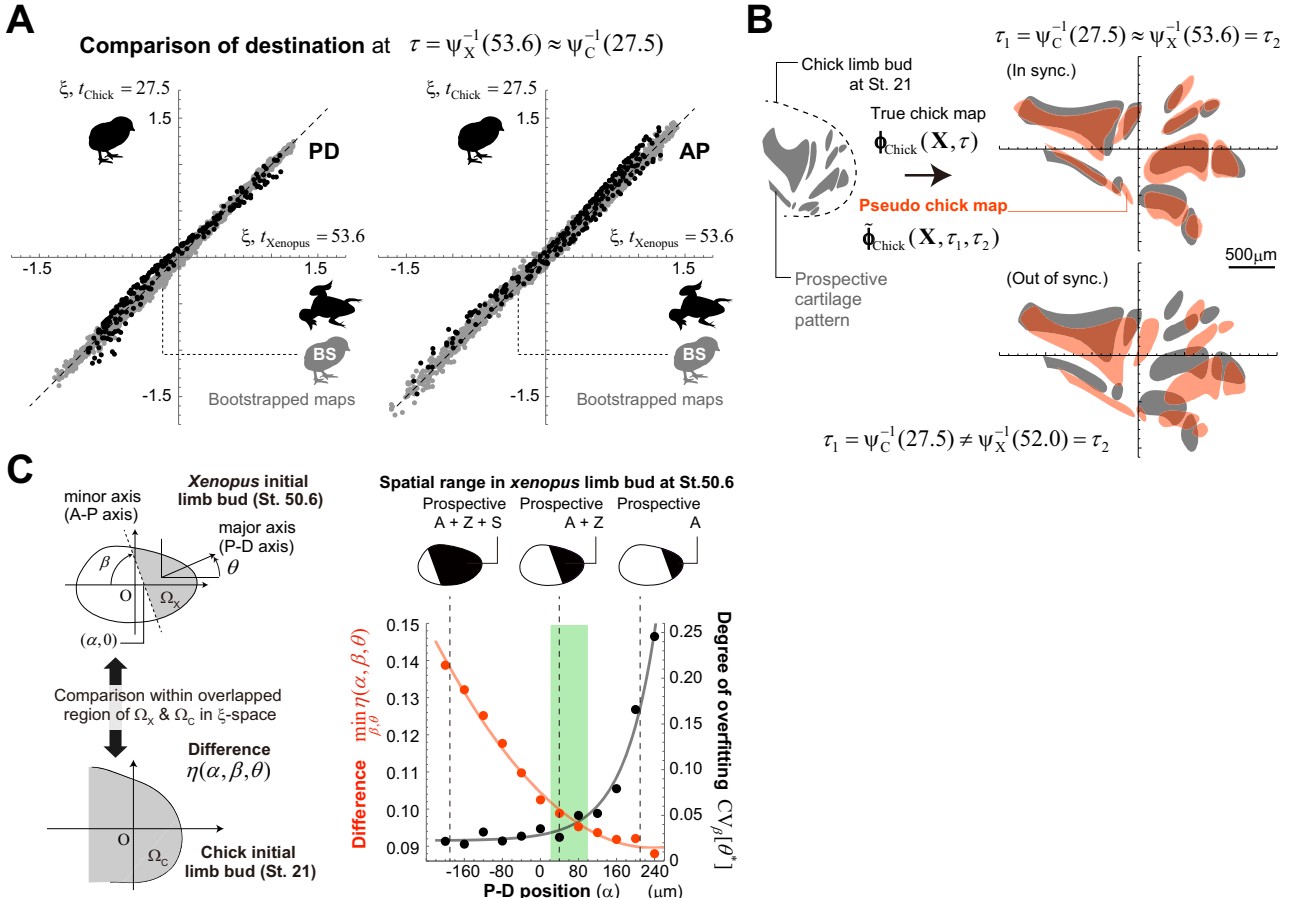

**Fig. 5 | Quantitative comparison of tissue dynamics in τ-ξ spacetime. A** (The black dots) Interspecies comparison of the destination of each cell at a common time $\tau = \psi_X^{-1}(53.6) \approx \psi_C^{-1}(27.5)$; P-D coordinate (left) and A-P coordinate (right). The gray dots indicate the variability in cell destinations within the same species (chick), which was evaluated using the bootstrapped tissue deformation maps as a reference against the difference between species. **B** Results of applying true and pseudo chick limb deformation maps to prospective skeletal patterns at $t_{Chick} = 21$. When the developmental clocks are synchronized and the state in the **ξ**-space at the same common time ($\tau = \tau_1 = \tau_2 = \psi_C^{-1}(27.5) \approx \psi_X^{-1}(53.6)$) is swapped between chick and *Xenopus*, the mapped skeletal patterns were very well matched, indicating that the rescaled tissue dynamics are conserved across species. In contrast, when the developmental clocks are out of synchronization, or the rescaled cell positions in *Xenopus* dynamics at a different time point ($\tau = \tau_1 \neq \tau_2$) are used to calculate the pseudo chick map, both mapped cartilage patterns are significantly different (see also Supplementary Fig. 6). **C** Comparison of the rescaled tissue dynamics within different regions of the limb buds between *Xenopus* and chick. The consistency of the rescaled dynamics between species holds only for anatomically corresponding regions (green region; see the text and Methods for details). Source data are provided as a Source Data file.

representation of positional information in the **ξ**-space for the regions from which anatomical structures are formed is quantitatively conserved across species having homologous organs of different sizes and shapes.

Thus, in the τ-**ξ** coordinate system, not only tissue dynamics but also expression dynamics of some typical marker genes are also conserved, further supporting the importance of the τ-**ξ** representation of dynamics in understanding the design of organ morphogenetic processes beyond individual species.

## Discussion

In the present study, we performed a quantitative comparative analysis of chick and *Xenopus* limb morphogenetic processes to address the interspecies geometric relationship in tissue dynamics during development. To compare the dynamics in tissues of different sizes/shapes and developmental rates, we proposed a spatial coordinate rescaled by the average dynamics (**ξ**-coordinate) and the common clock (τ-coordinate). The τ-**ξ** representation of tissue dynamics was shown to be conserved between these species, at least from the early stages of limb development through the phase when basic digit patterning has been established.

Based on the results of our analysis, we propose a new class of scaling. As described earlier, previous morphometric studies have revealed that the shape or spatial arrangement of landmarks in some animals can mirror each other through relatively simple transformations. In particular, if the transformations are linear or affine, they are regarded as scaling operations for shape. Denoting the (regular) matrix representing such transformations as **M**, and by decomposing it into a stretch operation $\mathbf{U}\ (= (\mathbf{M}^T\mathbf{M})^{1/2})$ and a rotation **R** (i.e., $\mathbf{M} = \mathbf{RU}$)[32], the eigenvectors of **U**, which are mutually orthogonal, give the axes of scaling along which the shape expands or shrinks (see Fig. 7A for specific examples of pure or shear scaling).

Beyond such scaling of developed organs/body shapes, our results demonstrate that the developmental process itself, i.e., internal tissue dynamics that determine organ morphology, can also scale between species in the following sense. It has been revealed that, behind the chick and *Xenopus* limb development, there is a non-linear dynamics that generates a common cell flow, $\boldsymbol{\xi}(\boldsymbol{\xi}_0, \tau)$, in the τ-**ξ** space (Fig. 7B). Multiplying this common flow by the species-specific average growth pattern, which is a linear operator denoted by $\mathbf{L}_{Chick}(\tau)$ for chick or $\mathbf{L}_{Xenopus}(\tau)$ for *Xenopus*, we obtain the original tissue dynamics, i.e., deformation map $\boldsymbol{\phi}_{Chick}$ or $\boldsymbol{\phi}_{Xenopus}$, for each species under a fixed

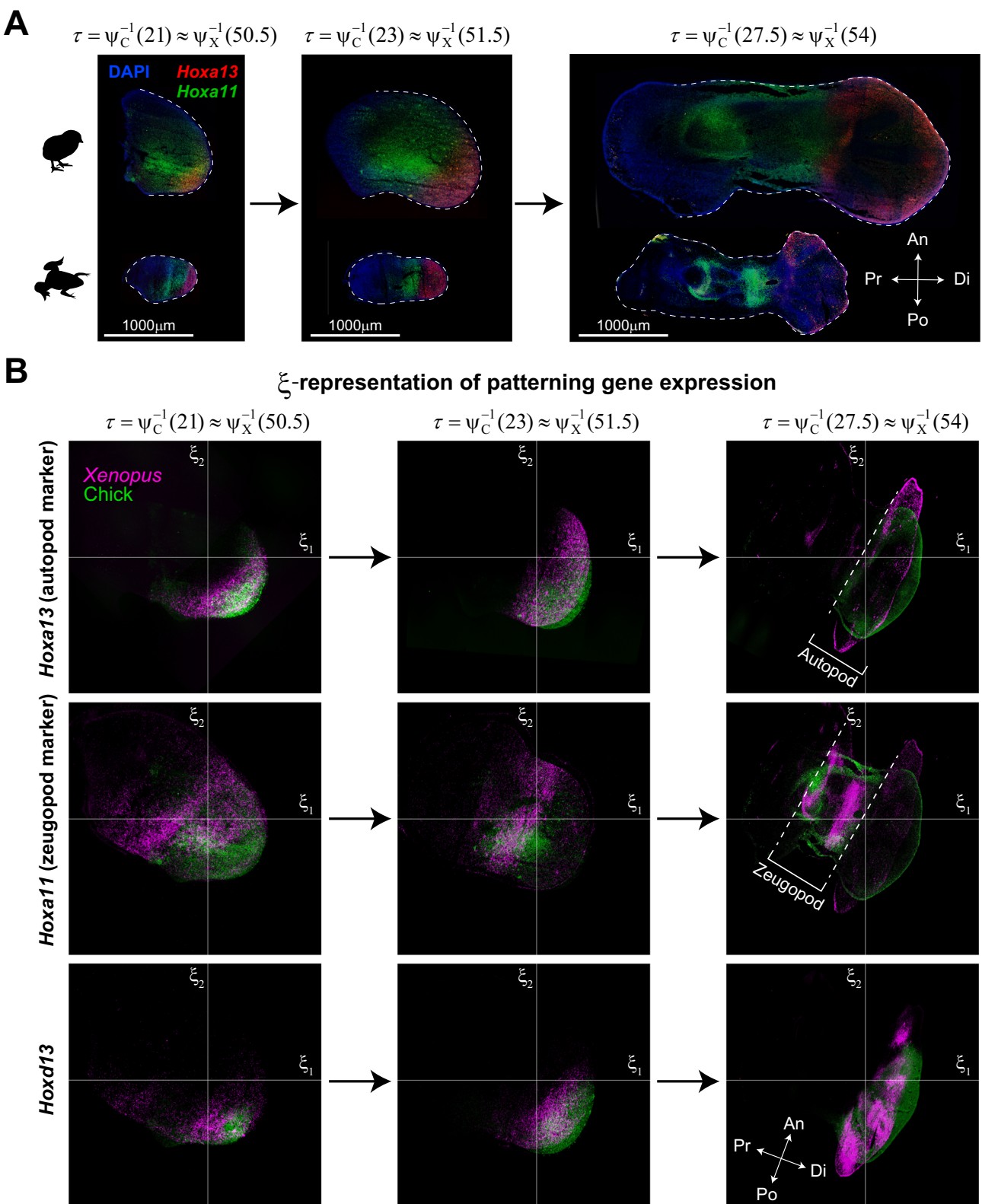

**Fig. 6 | Conservation of expression patterns for *Hox* genes in τ-ξ spacetime.**
**A** The expression patterns of *Hoxa13* and *Hoxa11* obtained by RNAscope assays at three different time points on the common clock (see Supplementary Fig. 7 for *Hoxd13* expression). Each experiment was independently repeated three times (at each time point) for each gene with similar results. **B** ξ-representation of the expression patterns of the *Hox* genes shows a high degree of similarity between species.

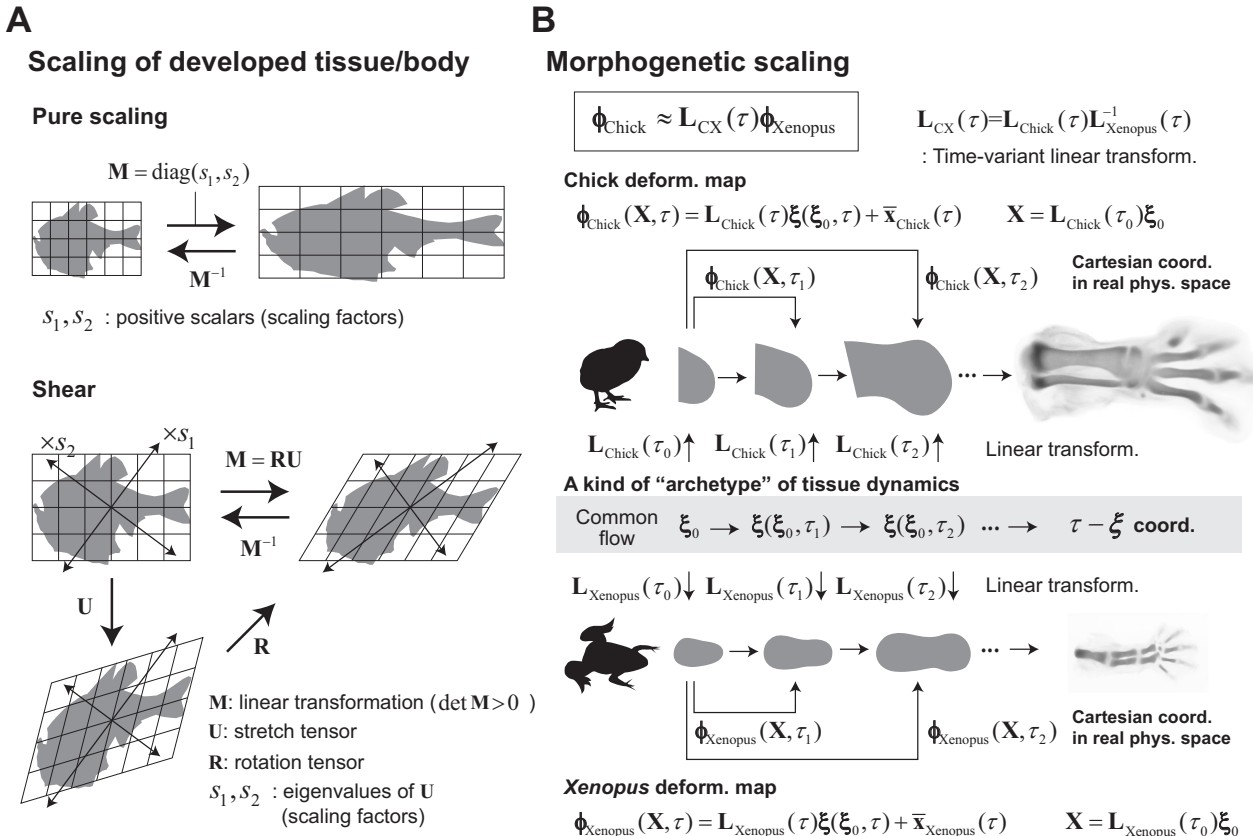

**A**

## Scaling of developed tissue/body

### Pure scaling

$\mathbf{M} = \mathrm{diag}(s_1, s_2)$

$s_1, s_2$ : positive scalars (scaling factors)

### Shear

$\times s_2$   $\times s_1$

$\mathbf{M} = \mathbf{RU}$

$\mathbf{M}^{-1}$

$\mathbf{U}$

$\mathbf{R}$

$\mathbf{M}$: linear transformation ($\det \mathbf{M} > 0$)
$\mathbf{U}$: stretch tensor
$\mathbf{R}$: rotation tensor
$s_1, s_2$ : eigenvalues of $\mathbf{U}$ (scaling factors)

**B**

## Morphogenetic scaling

$\boldsymbol{\phi}_{\mathrm{Chick}} \approx \mathbf{L}_{\mathrm{CX}}(\tau)\boldsymbol{\phi}_{\mathrm{Xenopus}}$     $\mathbf{L}_{\mathrm{CX}}(\tau) = \mathbf{L}_{\mathrm{Chick}}(\tau)\mathbf{L}_{\mathrm{Xenopus}}^{-1}(\tau)$
: Time-variant linear transform.

**Chick deform. map**

$\boldsymbol{\phi}_{\mathrm{Chick}}(\mathbf{X}, \tau) = \mathbf{L}_{\mathrm{Chick}}(\tau)\boldsymbol{\xi}(\boldsymbol{\xi}_0, \tau) + \bar{\mathbf{x}}_{\mathrm{Chick}}(\tau)$     $\mathbf{X} = \mathbf{L}_{\mathrm{Chick}}(\tau_0)\boldsymbol{\xi}_0$

$\boldsymbol{\phi}_{\mathrm{Chick}}(\mathbf{X}, \tau_1)$     $\boldsymbol{\phi}_{\mathrm{Chick}}(\mathbf{X}, \tau_2)$     **Cartesian coord. in real phys. space**

$\mathbf{L}_{\mathrm{Chick}}(\tau_0)\!\uparrow$   $\mathbf{L}_{\mathrm{Chick}}(\tau_1)\!\uparrow$   $\mathbf{L}_{\mathrm{Chick}}(\tau_2)\!\uparrow$   Linear transform.

**A kind of "archetype" of tissue dynamics**

Common flow   $\boldsymbol{\xi}_0 \rightarrow \boldsymbol{\xi}(\boldsymbol{\xi}_0, \tau_1) \rightarrow \boldsymbol{\xi}(\boldsymbol{\xi}_0, \tau_2) \cdots \rightarrow$   $\tau - \boldsymbol{\xi}$ coord.

$\mathbf{L}_{\mathrm{Xenopus}}(\tau_0)\!\downarrow$   $\mathbf{L}_{\mathrm{Xenopus}}(\tau_1)\!\downarrow$   $\mathbf{L}_{\mathrm{Xenopus}}(\tau_2)\!\downarrow$   Linear transform.

$\boldsymbol{\phi}_{\mathrm{Xenopus}}(\mathbf{X}, \tau_1)$     $\boldsymbol{\phi}_{\mathrm{Xenopus}}(\mathbf{X}, \tau_2)$     **Cartesian coord. in real phys. space**

*Xenopus* **deform. map**

$\boldsymbol{\phi}_{\mathrm{Xenopus}}(\mathbf{X}, \tau) = \mathbf{L}_{\mathrm{Xenopus}}(\tau)\boldsymbol{\xi}(\boldsymbol{\xi}_0, \tau) + \bar{\mathbf{x}}_{\mathrm{Xenopus}}(\tau)$     $\mathbf{X} = \mathbf{L}_{\mathrm{Xenopus}}(\tau_0)\boldsymbol{\xi}_0$

**Fig. 7 | Morphogenetic scaling and the archetype hypothesis. A** Simple transformations of shapes through diagonal and shear matrices as examples for the scaling of developed tissue/body shapes between species. In general, an arbitrary regular matrix **M** can be decomposed into the stretch operation **U** and the rotation **R** (i.e., **M** = **RU**), and thus linear transformations through regular matrices are regarded as a scaling operation of shapes along the direction of each eigenvector (of **U**). **B** Our results show that the rescaled tissue dynamics in the **ξ**-space are well conserved between chick and *Xenopus*. This flow can be regarded as an archetype, corresponding to the standard shape suggested by D'arcy Thompson's pioneering work. The tissue deformation map for each species is obtained by applying a species-specific time-variant linear transformation to this common flow. Consequently, the tissue dynamics for both species are also mapped, or scaled, to each other through time-variant linear transformations.

ordinary Cartesian coordinate system (Fig. 7B). Therefore, both maps align through the time-variant linear transformation $\mathbf{L}_{\mathrm{CX}}(\tau) = \mathbf{L}_{\mathrm{Chick}}(\tau)\mathbf{L}_{\mathrm{Xenopus}}^{-1}(\tau)$ as $\boldsymbol{\phi}_{\mathrm{Chick}} \approx \mathbf{L}_{\mathrm{CX}}(\tau)\boldsymbol{\phi}_{\mathrm{Xenopus}}$, which can be regarded as an extension of the classic scaling hypothesis for mature shapes. Moreover, the common cell flow in the **ξ**-space that is conserved between species is then regarded as a kind of archetype or normal form for organ specific dynamics, which may correspond to the standard shape suggested from the work of D'arcy Thompson (Fig. 1A).

As described in the Introduction, the trigger/timing of limb development are different between chick and *Xenopus*. Despite such qualitative differences, it is surprising that tissue dynamics are well conserved, which suggests that the evolution of limb morphogenesis is constrained by common physical processes as well as conserved signaling pathways and gene expression patterns across species. Clearly, such an interdisciplinary understanding is necessary to determine what an archetype of tissue dynamics is (including knowing if an archetype itself really exists). In the context of limb development, clarifying the physicochemical factors responsible for the nearly uniform elongation rate along the P-D axis and asymmetric growth and deformation along the A-P axis, as revealed in our analysis (Fig. 3C–H), will be an important clue to understanding the constraints. It should be noted that, in the context of evolution, it is important to exercise caution when using the term 'conservation.' Generally, the high similarities in tissue dynamics and associated genes make it plausible that these processes have been conserved without change from a common

ancestor. However, it is essential to remain open to the possibility that they may have independently evolved through convergence.

In the $\tau$-**ξ** coordinate system, the spatiotemporal expression patterns of *Hox* genes essential for limb development were also found to be conserved, demonstrating the importance of studying issues related to the coding or molecular representation of positional information in the $\tau$-**ξ** spacetime. Spatially, positions in the rescaled space (**ξ**), rather than those on an absolute scale, would be encoded as gene expression vectors in a manner common to different species. Changes in tissue size (or more precisely, average deformation $\mathbf{L}(\tau)$) are cancelled out in **ξ**-space, which allows direct comparisons of spatial representations within a tissue at different time points. This perspective will be crucial for advancing the previous theoretical frameworks on positional-information coding in static fields[33,34] to encompass dynamic scenarios.

With respect to time ($\tau$), we propose a method for achieving synchronization of developmental time between species based on geometrical information, i.e., cell flow under the **ξ**-coordinate system. The method has a limitation that it functions effectively only when the transformation of time coordinates between species adheres to the conditions of being both bijective and continuous. These conditions can be compromised when significant interspecies differences in the flow arise; in the case of chick and *Xenopus* limb development, (geometrical) synchronization was not possible after the phase when basic digit patterning was established. We should note that this does not necessarily imply only a negative aspect, because the method

conversely could also serve to detect the emergence of interspecies differences in tissue dynamics, as shown above.

Recent advances in spatial transcriptome technology have made it possible, in principle, to compare the expression patterns of the entire sets of genes during developmental processes[35,36]. Genes whose expression patterns in the $\tau$-$\xi$ spacetime are conserved among species are potential candidate genes responsible for encoding positional information, while non-conserved genes would likely be more relevant for determining species differences. In addition, utilizing the similarity of gene expression patterns between different species can provide an alternative approach to synchronize developmental time other than the one based on the tissue dynamics proposed here. It would be interesting to determine whether the times synchronized by both of those methods match, and whether there is a critical time point at which species diversity in both tissue and gene-expression dynamics begins to emerge.

In this study, we decomposed a tissue deformation map as the product of the average growth $\mathbf{L}(\tau)$, that determines species-specific tissue size and aspect ratio at time $t$, and the rescaled tissue dynamics $\xi(\xi_0, \tau)$, and focused on similarities in the latter. However, how the former, species-specific tissue size, is determined remains a critical issue because it not only satisfies pure scientific interest but also is closely linked to applied research. For example, current organoids can reproduce the differentiated states of individual organs, but remain considerably distant from being fully functional, mature organs of adequate size[37]. Identifying the factors that determine interspecies differences in the size of homologous organs will introduce the possibility of regulating the size of immature organoids. A group of genes associated with species differences whose expression patterns are not conserved in the $\tau$-$\xi$ coordinate system will provide clues to elucidate the molecular mechanisms responsible for organ size determination.

It will be a great challenge to determine the extent to which the rescaled dynamics conserved in chick and *Xenopus* are universal among species, and whether specific conserved dynamics are also present in other organs besides the limb. To define the $\tau$-$\xi$ coordinate system, a tissue deformation map is necessary, and obtaining such a map for various species/organs is a current limitation. Especially, since three-dimensional (3D) species-specific features of organ morphology become more pronounced in the later stages of development, 3D map reconstruction would be critical. In the context of limb development, at stages following the establishment of the skeletal patterning (e.g., after St. 30 in the chick case), the shapes of all digits become more distinct, while simultaneously assuming a more 3D arrangement relative to each other. Thus, extending the analysis to a 3D analysis will be essential for a deeper understanding of species differences in tissue dynamics. Thus, it will be necessary to develop a high-throughput method for reconstructing (preferably 3D) tissue deformation dynamics from imaging data. In the future, if we can obtain information on tissue dynamics over a wide time range for various species/organs, it will be possible to discuss the existence of an archetype or a normal form for each organ (i.e., species-independent, organ-specific, conserved rescaled dynamics) and it is expected that our fundamental understanding of design principles for morphogenetic dynamics will be dramatically improved. It will also be exciting if we can address another interesting question around whether the common clock defined by the tissue dynamics of one organ coincides with the clock defined by that of another organ, i.e., whether there is a common clock consistent throughout the embryonic body.

## Methods

### Ethical treatment and manipulation of animals

Experiments on Xenopus limb development were conducted at the National Institute of Basic Biology (NIBB), Tohoku University, and RIKEN Center for Biosystems Dynamics Research (RIKEN BDR). All procedures and protocols were approved by the Institutional Animal Care and Use Committee of NIBB. RIKEN and Tohoku University (and also Japanese domestic law, Act on Welfare and Management of Animals) exempt studies involving amphibians from requiring IRB approval, however, all experiments at RIKEN BDR and Tohoku University were performed in accordance with NIBB's ethics guidelines.

Wild-type *Xenopus laevis* tadpoles used for RNAscope assays were purchased from a domestic animal vendor. The hsp70-GFP F4 Tg *Xenopus* line containing GFP under the control of the hsp70 promoter[38] was gifted by Dr. Yumi Izutsu (according to the Xenbase nomenclature, this transgenic Xenopus laevis is named as *Xla.Tg (Xla.hsp70:eGFP)$^{Yumi}$*). The original plasmid used for the transgenesis of hsp70-GFP *Xenopus* was pHS1/EGFP[39]. We obtained the hsp70-GFP F5 Tg lines by crossing sexually mature F4 male frogs with WT female frogs, then used the F5 Tg *Xenopus* tadpoles for the IR-LEGO experiment. These Tg tadpoles were used for cell lineage tracing to reconstruct the tissue deformation maps. Tadpoles were reared at 22–23 °C in dechlorinated tap water and handled as previously reported[20].

Regarding chick embryos (wild type), we purchased fertilized chicken eggs from Shiroyama Farm and Inoue egg farm. The experiments comply with all ethical regulations because no licensing was required to study chick embryos at the stages used (HH20-HH30.5). The fertilized eggs were incubated in a humidified incubator at 38 °C to obtain embryos at the desired Hamburger and Hamilton (HH) stage and to perform the RNAscope assays.

In all experiments, the sex of tadpoles or chick embryos was not considered. This is because we focused on stages of embryonic development long before sexual maturity.

### Alcian blue and H&E staining

To visualize the progression of skeletal patterning, we performed H&E staining and Alcian blue staining; the former was used to visualize the spatial heterogeneity in cell density and the latter was used to identify cartilage differentiation. As described in the text, the timing of cartilage differentiation is delayed relative to cell density patterning.

For H&E staining of chick tissues, unfixed chick limb buds were embedded in 20% sucrose/OCT compound (Sakura Finetek Japan, 4583); 20% sucrose and OCT were mixed at a ratio of 1:4, tissues were then sectioned at 20-μm thickness on a cryostat. For *Xenopus* tissues, unfixed tadpole limb buds were embedded in OCT compound (Sakura Finetek Japan, 4583) and sectioned at 10-μm thickness on a cryostat. The sections were mounted on MAS-coated glass slides (Matsunami Glass Ind., Ltd.) and fixed for 30 min at −20 °C with 100% MeOH, then treated with 100% isopropanol for 1 min at RT followed by drying. H&E staining was performed by incubating sections in hematoxylin [Sigma-Aldrich, 51275] for 7 min, bluing buffer [Dako, CS70230-2] for 2 min, and an eosin solution containing 10% Eosin Y (Sigma-Aldrich, HT110216) and 90% 0.45 M Tris-Acetic Acid pH 6.0 for 1 min. Tissue sections were washed with water after each step, then the slides were dried and photographed using a Slide Scanner [Zeiss, AxioScan Z1].

For Alcian blue staining of chick tissues, chick limb buds were fixed in 10% formalin for several days at 4 °C, followed by washing with water for 1 day. For cartilage staining, the limb buds were dehydrated through a graded ethanol series (50%, 70%, 100%) over a period of three days, then incubated overnight at room temperature (RT) in Alcian blue solution containing 20% Alcian blue (Nacalai Tesque, 37154-44), 64% ethanol, and 16% acetic acid. After rehydration through a graded ethanol series (100%, 70%, 50%, water) over several hours, sections were treated with 1% KOH for transparency. For *Xenopus* limb buds, tissues were fixed in 4% PFA-PBS for 2 h at RT, followed by washing with water for 1 day; the epidermis was then peeled off to remove the pigment cells. After dehydration of the tissues through a series of ethanol (50% and 70%) for approximately 30 min each, the limb buds were further depigmented with a 3% $H_2O_2$-ethanol solution (3% $H_2O_2$, 70% ethanol) under light illumination for half a day, then again dehydrated in 100% ethanol overnight. After that, the limb buds

were incubated in an Alcian blue solution (same as above) overnight at RT, then in 95% ethanol for several days, followed by rehydration through a graded ethanol series (70%, 50%, and water). Finally, limb buds were treated with 1% KOH for transparency.

## Cell labeling by heat-shock induction of GFP expression and lineage tracing

As described in the main text, to obtain the cell lineage data necessary for reconstructing the tissue deformation maps, we labeled cells in the hindlimb bud of the above Tg animals by inducing EGFP expression using local irradiation with an infrared laser and the IR-LEGO system[19]. Since our aim was to compare the 2D deformation dynamics on the frontal plane at the mid D-V level with the maximal area of a limb bud in *Xenopus* with that previously reported for chick[14], we focused the irradiation within this plane (Fig. 2A). The heat-shock treatment was performed as described in our previous work[20]. The diameter of a single labeled spot on the plane was typically 20-30 μm, a size equivalent to a few cells. Although laser irradiation can be focused to a certain D-V level to some extent, each spot could become elongated in the D-V direction. However, we found that the in-plane shape of each spot was well maintained during our measurement time interval (around 24 h), and that even if we changed the focus of the microscope in the D-V direction, the spot position hardly changed on the plane. This means that the D-V dependence of the in-plane deformation is sufficiently small, at least in the measurement interval. It should be noted that tissue-level deformation is not calculated from the movement of cells within each spot, but from the change in the relative positions among the labeled spots. Therefore, if the relative positional changes between spots over the entire limb bud are reproducible, the deformation dynamics can be correctly reconstructed. In that sense, our data are highly reproducible among samples, and the resultant reconstructed tissue deformation dynamics are reliable. Since GFP expression was transient and the fluorescent signal weakened after several days due to dilution by cell division and protein degradation, it was difficult to analyze the lineages of labeled spots within a single embryo for a long period of time. Therefore, we measured the positional changes of the labeled spots in 11 individuals from different developmental stages (the measurement intervals of some individuals overlapped) and integrated the data from all individuals to cover the developmental period from initial limb bud stage to the stage in which digit pattern formation is clear (Supplementary Fig. 1C).

Staging of each individual was based on a previously proposed morphometric staging method[21]; briefly, digitized outlines of the limb buds were approximated using elliptic Fourier descriptors, and a continuous stage value, not discrete values as in traditional staging (e.g., 51 and 52), was assigned to each individual based on its coefficients. In this study, this morphometric stage was denoted by $t_{\text{Xenopus}}$. Each limb bud was resized to a previously determined typical value corresponding to its morphometric stage. The spatial coordinates of the fluorescently labeled spots were measured at 2 to 4 time points approximately every 24 h for each individual, and the coordinates at the timepoints between the measurements were obtained by linear interpolation. The period from $t_{\text{Xenopus}} = 50.6$ to $t_{\text{Xenopus}} = 54.4$ was divided into nine roughly equally spaced time intervals (each interval corresponded to a 0.4–0.5 stage increment), and data from the individuals included in each interval were integrated to reconstruct the tissue deformation map for that interval.

## Reconstruction of tissue deformation maps

Let $\mathbf{x} = \boldsymbol{\phi}_{\text{Xenopus}}^{(\alpha)}(\mathbf{X})$ be the tissue deformation map for $\alpha$-th time interval ($\alpha = 1, \cdots, 9$) during *Xenopus* hindlimb development, where $\mathbf{X}$ and $\mathbf{x}$ are the positional vectors of a cell or a piece of tissue before and after deformation during a focal time interval, respectively. As described above, from the measurement of the positional changes of the spots

fluorescently labeled by heat shock, datasets $(\mathbf{X}_i^{(\alpha)}, \mathbf{x}_i^{(\alpha)})$ ($i = 1, \cdots, N_\alpha$; $\alpha = 1, \cdots, 9$) were obtained where $N_\alpha$ is the number of the data point (i.e., the spot number) for the $\alpha$-th interval; ($N_1, N_2, N_3, N_4, N_5, N_6, N_7, N_8, N_9$) = (113, 185, 67, 79, 226, 333, 256, 129, 270). From these datasets, the maps for each interval were reconstructed using the Bayesian method as reported previously[13]. With the composite map $\mathbf{x} = \boldsymbol{\phi}_{\text{Xenopus}}(\mathbf{X}, t)$ obtained by integrating the maps for all intervals, it becomes possible to calculate the trajectory of each cell from the initial time point ($t_{\text{Xenopus}} = 50.6$) to the final time point ($t_{\text{Xenopus}} = 54.4$). The bootstrapped maps used in Fig. 5A and Supplementary Fig. 5 were obtained by applying the same Bayesian reconstruction method to the bootstrapped datasets generated from the original datasets above. For chick hindlimb development, we used a previously reported tissue deformation map $\mathbf{x} = \boldsymbol{\phi}_{\text{Chick}}(\mathbf{X}, t)$[14]. The only minor modification from the previous study was that we changed the staging method. In the previous study, a tissue deformation map was quantified for every 12-hour interval, and for each time point, we assigned the integer value of the Hamburger-Hamilton stage with a shape closest to that observed at the time point. In the present study, staging was performed using increments of 0.5 (not necessarily integers) based on incubation time. The amount of local deformation at time $t$ (with the initial configuration as the reference) around a cell initially located at $\mathbf{X}$ can be given by the deformation gradient tensor $\mathbf{F}(\mathbf{X}, t) = \partial \boldsymbol{\phi} / \partial \mathbf{X}$[32]. $\mathbf{L}(t)$ in Eq. 1 is the spatial average of this tensor for a given time.

## Representation of tissue deformation dynamics in the ξ-space

As described in the main text, we defined the rescaled position $\boldsymbol{\xi}$ by applying the inverse operation of the product of the initial centroid size of a tissue ($C_0$) and the spatial average of the deformation gradient tensor that represents local tissue growth under the ordinary Cartesian coordinate ($\bar{\mathbf{F}}(t)$); the product is denoted by $\mathbf{L}(t) \equiv C_0 \bar{\mathbf{F}}(t)$, which can be interpreted as multi-dimensional tissue size at time $t$. More specifically, $\mathbf{L}(t)$ and the rescaled position $\boldsymbol{\xi}(\boldsymbol{\xi}_0(\mathbf{X}), t)$ are given as follows:

$$\mathbf{L}(t) = C_0 \frac{\int_{\Omega_0} \mathbf{F}(\mathbf{X}, t) d\mathbf{X}}{\int_{\Omega_0} d\mathbf{X}}, \tag{5}$$

$$\boldsymbol{\xi}(\boldsymbol{\xi}_0(\mathbf{X}), t) = \mathbf{L}^{-1}(t)(\mathbf{x}(\mathbf{X}, t) - \bar{\mathbf{x}}(t)), \tag{6}$$

where $\Omega_0$ and $\mathbf{F}(\mathbf{X}, t)$ are the domain of the initial tissue and the deformation gradient tensor at time $t$ for the cell initially located at $\mathbf{X}$, respectively. $\boldsymbol{\xi}_0 \equiv \mathbf{X}/C_0$ is the $\boldsymbol{\xi}$-representation of the initial position $\mathbf{X}$, and $\bar{\mathbf{x}}(t)$ is the geometric tissue center at $t$. Without loss of generality, let $\bar{\mathbf{x}}(0) = \mathbf{0}$ and $\mathbf{x}(\mathbf{0}, t) = \mathbf{0}$ in the following analysis.

The cell trajectory in the $\boldsymbol{\xi}$-space represents the change in cell position relative to the entire tissue after canceling out the average growth. The velocity field that determines the cell trajectory is given as follows. Differentiating Eq. (1) by $t$,

$$\frac{D}{Dt}\boldsymbol{\xi}(\boldsymbol{\xi}_0, t) = \frac{\partial}{\partial t}\boldsymbol{\xi}(\boldsymbol{\xi}_0, t) = \mathbf{L}^{-1}(t)\frac{\partial}{\partial t}(\mathbf{x}(\mathbf{X}, t) - \bar{\mathbf{x}}(t)) + \frac{\partial \mathbf{L}^{-1}(t)}{\partial t}(\mathbf{x}(\mathbf{X}, t) - \bar{\mathbf{x}}(t))$$
$$= \mathbf{L}^{-1}(t)\frac{\partial}{\partial t}(\mathbf{x}(\mathbf{X}, t) - \bar{\mathbf{x}}(t)) - \mathbf{L}^{-1}(t)\bar{\mathbf{S}}(t)(\mathbf{x}(\mathbf{X}, t) - \bar{\mathbf{x}}(t)), \tag{7}$$

where $D/Dt$ is the material time derivative and we introduced $\bar{\mathbf{S}}(t)$, a kind of spatial average of the velocity gradient tensor, defined by:

$$\frac{\partial \bar{\mathbf{F}}(t)}{\partial t} = \bar{\mathbf{S}}(t)\bar{\mathbf{F}}(t). \tag{8}$$

When the initial tissue shape is a star-domain at least for the origin $\mathbf{0}$ (note that the limb buds for chick and *Xenopus* satisfy this condition) and tissue deformation is smooth (e.g., C$^2$-class), the deformation map

or current position of each cell is represented as:

$$\mathbf{x}(\mathbf{X},t) = \int_0^1 \mathbf{F}(\varepsilon\mathbf{X},t)\mathbf{X}\,d\varepsilon = \left(\int_0^1 \mathbf{F}(\varepsilon\mathbf{X},t)d\varepsilon\right)\mathbf{X}. \quad (9)$$

This equation may be easier to understand by looking at its component form in a 2D case.

$$
\begin{aligned}
x_i(\mathbf{P},t) &= x_i(\mathbf{P},t) - x_i(\mathbf{0},t) \\
&= \int_\Gamma \left(\frac{\partial x_i}{\partial X_1}, \frac{\partial x_i}{\partial X_2}\right) \cdot d\mathbf{X} \\
&= \int_0^1 \left(\frac{\partial x_i(\varepsilon\mathbf{P},t)}{\partial X_1}, \frac{\partial x_i(\varepsilon\mathbf{P},t)}{\partial X_2}\right) \cdot \mathbf{P}\,d\varepsilon \\
&= \int_0^1 (F_{i1}(\varepsilon\mathbf{P},t), F_{i2}(\varepsilon\mathbf{P},t)) \cdot \mathbf{P}\,d\varepsilon
\end{aligned} \quad (10)
$$

The first equal sign holds with the assumption $\mathbf{x}(\mathbf{0},t) = \mathbf{0}$. The second line represents the line integral along curve $\Gamma$ connecting $\mathbf{0}$ and a focal point $\mathbf{P}$. The equality is satisfied when $\partial^2 x_i/\partial X_1 \partial X_2 = \partial^2 x_i/\partial X_2 \partial X_1$ holds (i.e., smooth deformation). Based on the star-domain assumption, a line segment connecting $\mathbf{0}$ and $\mathbf{P}$ is adopted as $\Gamma$ in the third line. A change of variables $\mathbf{X} = \mathbf{P}\varepsilon$ was also performed. The last line shows the representation using the components of the deformation gradient tensor.

Using Eq. 9, Eq. 7 can be transformed as follows:

$$\frac{\partial}{\partial t}\boldsymbol{\xi}(\boldsymbol{\xi}_0,t) = \mathbf{L}^{-1}(t)(\bar{\mathbf{S}}_{\mathbf{X}}(t) - \bar{\mathbf{S}}(t))\mathbf{x}(\mathbf{X},t) - \mathbf{L}^{-1}(t)\left(\frac{\partial\bar{\mathbf{x}}(t)}{\partial t} - \bar{\mathbf{S}}(t)\bar{\mathbf{x}}(t)\right), \quad (11)$$

where we again introduce a type of velocity gradient tensor $\bar{\mathbf{S}}_{\mathbf{X}}(t)$ that is defined by,

$$\frac{\partial}{\partial t}\left(\int_0^1 \mathbf{F}(\varepsilon\mathbf{X},t)d\varepsilon\right) = \bar{\mathbf{S}}_{\mathbf{X}}(t)\left(\int_0^1 \mathbf{F}(\varepsilon\mathbf{X},t)d\varepsilon\right). \quad (12)$$

$\bar{\mathbf{S}}_{\mathbf{X}}(t)$ can be interpreted as a spatial average of the velocity gradient tensor on the line segment that connects $\mathbf{0}$ to the focal point $\mathbf{X}$. Then, substituting $\mathbf{L}(t)\boldsymbol{\xi}(\mathbf{X},t) + \bar{\mathbf{x}}(t)$ for $\mathbf{x}(\mathbf{X},t)$ gives

$$
\begin{aligned}
\frac{\partial}{\partial t}\boldsymbol{\xi}(\boldsymbol{\xi}_0,t) &= \mathbf{L}^{-1}(t)(\bar{\mathbf{S}}_{\mathbf{X}}(t) - \bar{\mathbf{S}}(t))\mathbf{L}(t)\boldsymbol{\xi}(\boldsymbol{\xi}_0,t) - \mathbf{L}^{-1}(t)\left(\frac{\partial\bar{\mathbf{x}}(t)}{\partial t} - \bar{\mathbf{S}}_{\mathbf{X}}(t)\bar{\mathbf{x}}(t)\right) \\
&= \mathbf{L}^{-1}(t)(\bar{\mathbf{S}}_{\mathbf{X}}(t) - \bar{\mathbf{S}}(t))\mathbf{L}(t)\boldsymbol{\xi}(\boldsymbol{\xi}_0,t) - \mathbf{v}_g(\boldsymbol{\xi}_0(\mathbf{X}),t).
\end{aligned} \quad (13)
$$

$\mathbf{v}_g(\boldsymbol{\xi}_0(\mathbf{X}),t)$ represents the effect of the dynamics of the geometric center of the tissue. Given the assumption that $\bar{\mathbf{x}}(0) = \mathbf{0}$, when the change in its position is small, this term can be ignored. In the 1D case, Eq. 13 takes a simpler form,

$$\frac{d}{dt}\xi(\xi_0,t) = (\bar{S}_X(t) - \bar{S}(t))\xi(\xi_0,t) - v_g(\xi_0,t), \quad (14)$$

and the initial length $l_0$ may be used instead of the centroid size $C_0$.

### Common clock T and synchronization between species

As described in the main text, in this study we introduced a common clock $\mathbf{T}$ to synchronize the developmental time between species (Fig. 4A). The common clock is defined as a directed curve or one-dimensional manifold. $\mathbf{T}$ is an abstract concept and does not need a concrete scale. Each time on the common clock corresponds to a point $\tau$ on $\mathbf{T}$ and is quantified as the traditional developmental stage for each species (denoted by $t_{\text{Species}}$) through species-specific mapping $\psi_{\text{Species}}(\tau)$; in other words, time coordinates are defined for each species (Fig. 4A). Synchronizing the developmental time

between the two species ($\alpha$ and $\beta$) is to correspond their stages with each other via the common clock $\mathbf{T}$, i.e., to identify $t_\alpha = \psi_\alpha(\tau)$ and $t_\beta = \psi_\beta(\tau)$ for each common time $\tau$ and to obtain the composite map $t_\beta = \psi_\beta \circ \psi_\alpha^{-1}(t_\alpha)$ or $t_\alpha = \psi_\alpha \circ \psi_\beta^{-1}(t_\beta)$. Geometrically, the composite map represents a (local) coordinate transformation on the manifold $\mathbf{T}$.

In Fig. 4B, to obtain the composite map $\psi_C \circ \psi_X^{-1}$ or $\psi_X \circ \psi_C^{-1}$ that relates the chick and *Xenopus* developmental stages, we measured the distance ($\delta$) of each pair of developmental stages for chick and *Xenopus* ($t_{\text{Chick}}, t_{\text{Xenopus}}$) using the spatial average of the difference in cell positions at $t_{\text{Chick}}$ and $t_{\text{Xenopus}}$ on the trajectories for both species with the same initial value $\boldsymbol{\xi}_{0,i}$ (heatmap in Fig. 4B):

$$\delta(t_{\text{Xenopus}}, t_{\text{Chick}}) = \left\langle\left|\boldsymbol{\xi}_{\text{Xenopus}}(\boldsymbol{\xi}_{0,i}, t_{\text{Xenopus}}) - \boldsymbol{\xi}_{\text{Chick}}(\boldsymbol{\xi}_{0,i}, t_{\text{Chick}})\right|\right\rangle_{i\in\Omega}, \quad (15)$$

where $\Omega$ is the overlapping area of the prospective autopod and zeugopod regions within the limb buds of both species in the $\boldsymbol{\xi}$-space (see the colored area in Fig. 3C). Based on this, for each *Xenopus* stage $t_{\text{Xenopus}}^k$ ($k$: discretized time points) the closest chick stage (denoted by $t_{\text{Chick}}^{k,\text{XtoC}}$) was assigned (white/gray points and curve in Fig. 4B), which defines the composite map $\psi_C \circ \psi_X^{-1}$, and vice versa (black/gray points and curve):

$$t_{\text{Chick}}^{k,\text{XtoC}} = \arg\min_t \left\langle\left|\boldsymbol{\xi}_{\text{Xenopus}}(\boldsymbol{\xi}_{0,i}, t_{\text{Xenopus}}^k) - \boldsymbol{\xi}_{\text{Chick}}(\boldsymbol{\xi}_{0,i}, t)\right|\right\rangle_{i\in\Omega}, \quad (16a)$$

$$t_{\text{Xenopus}}^{k,\text{CtoX}} = \arg\min_t \left\langle\left|\boldsymbol{\xi}_{\text{Xenopus}}(\boldsymbol{\xi}_{0,i}, t) - \boldsymbol{\xi}_{\text{Chick}}(\boldsymbol{\xi}_{0,i}, t_{\text{Chick}}^k)\right|\right\rangle_{i\in\Omega}. \quad (16b)$$

Assuming that the common clock $\mathbf{T}$ runs smoothly (i.e., $\mathbf{T}$ is at least a topological manifold), the composite maps should be bijective and continuous (i.e., homeomorphisms). In our case, those conditions are satisfied up to around $t_{\text{Xenopus}} = 53.5$ and around $t_{\text{Chick}} = 27$. In particular, within or near the ranges $t_{\text{Xenopus}} \in [50.6, 53.5]$ and $t_{\text{Chick}} \in [21, 27]$ both stages correspond linearly.

In Fig. 5C, we compared cell trajectories in the $\boldsymbol{\xi}$-space within the whole chick limb bud ($\Omega_C$) that will form the autopod and zeugopod (Fig. 1B) and within different regions of the *Xenopus* limb bud ($\Omega_X$). The region $\Omega_X$ to be compared was defined by three parameters: $\alpha$, $\beta$, and $\theta$ (Fig. 5C). For each parameter set ($\alpha$, $\beta$, $\theta$), $(t_{\text{Xenopus}}^k, t_{\text{Chick}}^{k,\text{XtoC}}(\alpha, \beta, \theta))$ determines the correspondence of developmental stages between the two species. Then, the differences in tissue dynamics were evaluated as the temporal averages of $\delta(t_{\text{Xenopus}}^k, t_{\text{Chick}}^{k,\text{XtoC}})$ within the period $t_{\text{Xenopus}}^k \in [50.6, 54.4]$, which was denoted by $\eta$:

$$\eta(\alpha, \beta, \theta) = \left\langle\delta(t_{\text{Xenopus}}^k, t_{\text{Chick}}^{k,\text{XtoC}}(\alpha, \beta, \theta))\right\rangle_k. \quad (17)$$

The spatial range along the P-D axis of $\Omega_X$ is determined by $\alpha$. We calculated $\min_{\beta,\theta}\eta(\alpha, \beta, \theta)$ for each $\alpha$ and found that its value became smaller as $\Omega_X$ was increasingly restricted to the distal side (red curve in Fig. 5C (right)). When the range of $\Omega_X$ is too small, its shape in the $\boldsymbol{\xi}$-space deviates greatly from the chick limb bud shape, reducing the overlap region $\Omega$ between the two species; by correctly choosing $\beta$ and $\theta$, it is possible to closely match the *Xenopus* cell trajectory to that of chick within $\Omega$. This is a type of overfitting and does not guarantee robustness of the conclusion relative to the parameter values. Since the tissue orientation $\theta$ that minimizes the difference in tissue dynamics $\eta$ (denoted by $\theta^*(\alpha, \beta) = \arg\min_\theta \eta(\alpha, \beta, \theta)$) was sensitively dependent on $\beta$, we measured the degree of overfitting by its coefficient of variation:

$$\text{CV}_\beta[\theta^*] = \sqrt{\text{Var}_\beta[\theta^*]}/\text{E}_\beta[\theta^*]. \quad (18)$$

This value depended monotonically on $\alpha$ and became larger as the *Xenopus* limb bud region $\Omega_X$ was restricted to the distal side (Fig. 5C (right)). Lower values of $CV_\beta[\theta^*]$ and $\min_{\beta,\theta}\eta(\alpha,\beta,\theta)$ were compatible only when $\alpha$ assumed the values in which $\Omega_X$ corresponded to the prospective autopod and zeugopod (green region in Fig. 5C [right]). This also supports the idea that the consistency of the rescaled dynamics between species holds only for anatomically corresponding regions.

As described in the text, the values for the distance of each pair from the developmental stages of each species, $\delta(t_{\text{Xenopus}}, t_{\text{Chick}})$, depends on the choice of the initial time points. Here, we examined the following two cases: First, the initial time for *Xenopus* was fixed at $t_{\text{Xenopus}} = 50.6$, and four initial times, $t_{\text{Chick}} = 20, 21, 23, 24$, were tested in the calculations for the chick cell trajectories in the $\boldsymbol{\xi}$-space. For each initial value for chick, $\min_\theta[\eta(\alpha,\beta,\theta)]$ was calculated for 16 different parameter sets $(\alpha,\beta)$, where values corresponding to the position around the boundary between the prospective zeugopod and stylopod were chosen for $\alpha$, and values corresponding to the orientation around the A-P axis were chosen for $\beta$. As a result, $t_{\text{Chick}} = 21$ was the best fit for the initial time corresponding to $t_{\text{Xenopus}} = 50.6$ (Fig. 4E). In Fig. 4E, the medians and standard deviations are shown for the 16 sets of $(\alpha,\beta)$. The standard deviations are not large, showing the robustness of the values of $\min_\theta[\eta(\alpha,\beta,\theta)]$. Second, we performed a similar analysis for the case in which the initial time for chick was fixed at $t_{\text{Chick}} = 21$. The five initial times, $t_{\text{Xenopus}} = 50.6, 51.1, 51.6, 52.0, 52.4$ were tested in the calculations for the *Xenopus* cell trajectories in the $\boldsymbol{\xi}$-space. For each initial value for *Xenopus*, $\min_\theta[\tilde{\eta}(\alpha,\beta,\theta)]$ was calculated for the same 16 parameter sets for $(\alpha,\beta)$ as above, where $\tilde{\eta}$ is defined as follows:

$$\tilde{\eta}(\alpha,\beta,\theta) = \left\langle \delta(t^k_{\text{Chick}}, t^{k,C\text{toX}}_{\text{Xenopus}}(\alpha,\beta,\theta)) \right\rangle_k, \tag{19}$$

$$t^k_{\text{Chick}} \in [21, 30.5].$$

Consequently, the best agreement of trajectories between the two species was found at $t_{\text{Xenopus}} = 50.6$ (51.1 was equally good) (Supplementary Fig. 4). Based on these results, we chose $t_{\text{Xenopus}} = 50.6$ and $t_{\text{Chick}} = 21$ as the initial time combination. Note that the aspect ratios (P-D vs A-P) of the limb buds (for the prospective autopod and zeugopod) of both species are similar at $t_{\text{Xenopus}} = 50.6$ and $t_{\text{Chick}} = 21$.

### RNAscope assays
Tissue sections from *Xenopus* and chick hindlimb buds were prepared in the same way described for H&E staining. For *Xenopus* tissues, fluorescence in situ hybridization was carried out using the RNAscope Fluorescent Multiplex Reagent kit (Advanced Cell Diagnostics, #320850) to visualize the spatiotemporal expression of *Xl-hoxA11.L* (#850291), *Xl-hoxA13.L* (#850301), *Xl-hoxD13.L* (#891621) following the manufacturer's instructions. Briefly, sections were fixed for 15 min at 4 °C with 4% PFA/PBS then treated with protease IV for 30 min at RT. Probes were hybridized for 2 h at 40 °C in a HybEZ II oven. The sections were counterstained with DAPI (IBC, AR-6501-01) and imaged with a Digital Slide Scanner (Axio Scan Z1, Zeiss) and confocal microscope (LSM880, Zeiss). For chick tissues, the same RNAscope reagent kit was used to visualize the spatiotemporal expression of *Gg-HOXA11* (#893441), *Gg-HOXA13* (#893451), *Gg-HOXD13* (#893461), following the manufacturer's instructions. Briefly, the tissue sections were fixed for 15 min at 4 °C with 4% PFA/PBS, boiled in 1× Target Retrieval Reagent (Advanced Cell Diagnostics, #322000) for 5 min, then treated with protease III for 30 min at 40 °C. Probes were hybridized for 2 h at 40 °C in a HybEZ II oven. Sections were counterstained with DAPI (IBC, AR-6501-01) and imaged with a Digital Slide Scanner (Axio Scan Z1, Zeiss) and confocal microscope (LSM880, Zeiss).

### Statistics and reproducibility
We estimated the required number of labeled cells to reconstruct deformation maps of the Xenopus hindlimb for each time interval based on our previous studies related to the reconstruction of deformation maps of the chick hindlimb[13,14]. The residual errors in map estimation were several tens of micrometers, indicating that Bayesian regression worked effectively. More precisely, the mean residual errors were 18.4, 14.6, 4.5, 18.6, 10.5, 25.3, 30.0, 13.8, and 42.4 micrometers for each of the nine time intervals. In addition, we verified the similarity of cell trajectories across embryos during overlapping measurement periods.

### Reporting summary
Further information on research design is available in the Nature Portfolio Reporting Summary linked to this article.

## Data availability
All source data are provided with this paper in the Supplementary information. Source data are provided with this paper.

## Code availability
The code used for calculating the $\boldsymbol{\xi}$ coordinates as proposed in this paper is available as a Mathematica file in the Supplementary information (import data is also included).

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

## Acknowledgements

This work was supported by JST PRESTO Grant Number JPMJPR1779 and JST CREST Grant Number JPMJCR2025 to Y.M., and also supported by the NIBB Collaborative Research Program to H.Y. (16-509, 17-505) and to Y.M. (16-521, 17-512). We thank Dr. Yumi Izutsu for providing the hsp70-GFP Tg *Xenopus* frogs. We thank the Center of Research Instruments, Graduate School of Biomedical Engineering, Tohoku University for the use of the Nikon A1R MP laser-scanning confocal microscope. We would also like to thank Kaori Niimi, technical staff at the RIKEN Center for Biosystems Dynamics Research (BDR), for assisting with the H&E and Alcian blue staining, and Daisuke Ohtsuka, a senior researcher at the RIKEN BDR, for helpful comments.

## Author contributions

Y.M. designed this study, performed all the data analysis and mathematical formulations, and wrote the paper. A.K.-K. performed all experiments using the *Xenopus laevis* tadpoles including cell labelling by IR-LEGO, cell lineage tracing using two-photon microscopy, and RNAscope assays. S.-W.L. processed the imaging data. T.S. performed the lineage tracing experiments using chick embryos. H.Y. and Y.K. provided guidance on techniques for inducing gene expression by heat shock using the IR-LEGO system. K.T. supported the setup of the IR-LEGO system and provided very helpful comments on the experiments as an expert in developmental limb studies. All authors discussed the results.

## Competing interests

The authors declare no competing interests.
