## [Peer Review File · Nature Communications]

An archetype and scaling of developmental tissue dynamics across speciesREVIEWER COMMENTS

Reviewer #1 (Remarks to the Author):

This is an interesting study attempting to find a common morphometric plan underlying the development of hindlimbs in two vertebrate model organisms which differ in size, shape and developmental timing. It combines mathematical modelling with cell tracking in vivo, and builds on a previous study of the chicken hind limb bud. The study uses a very clever method of labelling many small groups of cells in the mesenchyme along with mathematical modelling which supports an “archetype” for stage 52 to 54. This stage interval is about 5 days of *Xenopus* tadpole development (starting 9 days after the buds appear, and finishing as digits begin to condense). At these stages the mesenchyme anterior-posterior axis is well established, and cells fated to form the stylopod, zeugopod and autopod can be identified (Tschumi 1957). Effectively, the study and model covers the period when zeugopodal cartilage elements condense and when digits and interdigits are being defined in the autopod. The latter has been recently shown to be recapitulated in in vitro culture in mice, indicating self-organising ability of the limb bud mesenchyme (Fuiten et al PMID: 36994104). The model is backed up with some nice RNAscope data of anterior posterior marker hox genes. There are also of course more interdigital regions in frog hindlimb (5 digits) vs. chick hindlimb (4 digits) and I can't see this reflected in the model or discussed. Major: The paper would benefit from some more background into the model to be more accessible to biologist readers. The discussion is brief and includes no citations, so is more a conclusion. I think at the least it would be good to address the limitations of this model in terms of the stages it covers, and perhaps to expand the discussion to include reference to the work of others – e.g. What is known about cell division rate and orientation in these models, or in other vertebrates, and how would this direct morphogenetic growth in a comparable way? Additionally I suggest changes that I think would assist with broad readership appeal as “minor” below.

Minor:

Throughout: species names in italics

Abstract: include full species names for both chicken and *Xenopus*

line 71 I am not familiar with the exact meaning of “tissue deformation” in this context, and think it would be more accessible to wider readership if this was briefly explained.

Line 84: similarly, what is a tissue deformation map?

Line 114: can you explain why “on the dorsal ventral boundary plane” and what this means? Is the grid focussed somehow so that epithelial cells are not triggered to express GFP? The image in figure 2A (and S1) is really nice but I cannot tell if it is mesenchymal only – were two lasers used to focus on a single spot, and only where they meet the heat shock promoter activates?

Line 120/121 state the Nieuwkoop and Faber stages used?

Line 135: *Xenopus* limb bud stages are points on a continuum, but how did you accurately subdivide each stage in to 10th stages?

Line 136 state HH stages

Line 139 citations are needed to support this “*Xenopus* limb buds include the prospective autopod (toe-to-ankle), zeugopod (lower leg), and stylopod (upper leg) regions. In contrast, chick limb buds contain mainly the former two, while the stylopod is embedded in the trunk.

Line 173 gene names in italics

Line 229 “we can intuitively see...” please explain what we are supposed to be seeing because I cannot make this conclusion from the coloured vectors in figure 3C and D. Are there even interdigits present at stage 52? Maybe this refers to another figure?

Paragraph starting line 236: what criteria did you use/match to set the clock for each

organism (e.g cartilage condensation, gene expression?)

Line 487 H₂O₂

Line 499 I'm still not understanding what you mean by dorsal ventral boundary: the mesenchyme does not really have such a boundary (unlike the epithelial AER?) – does this mean cells were labelled in the centre relative to the DV axis, where *lmx1b* positive and negative cells meet? Could you perhaps illustrate this better with a figure?

Line 669 Gene names in italics

Figure 1: I find this legend to be a bit overlong, it should really be re-written to just describe what is shown and the rest can be incorporated into the text? In particular, having a study aim in the legend seems out of place.

Figure 2 The *Xenopus* A-P axis is posturally inverted: *shh* marking the ZPA clearly shows this for the stages of this study: Endo et al 1997 PMID: 9186057, Keenan and Beck 2016 PMID: 26404044, Wang et al 2015 PMID: 26527308. I can see you've inverted the limbs in this figure to account for this but it should be stated.

I'm not clear on how the maps are made- 1D or is information captured from multiple layers and stacked? If the former then how do you account for cells that move or divide dorsoventrally in the mesenchyme?

Figure 3 rather than using left and right, for C and D maybe label all the panels? In the legend, the final statement "Note that the dynamics for the prospective autopod and zeugopod are compared" at the end does not seem to obviously relate to anything shown in the figure?

Reviewer #2 (Remarks to the Author):

ms review

NCOMMS-23-19472-T

The goal of the paper is to explore whether there are conserved 'archetypal' growth dynamics in organs detectable among species. The authors produce tissue deformation maps through developmental stages of chick and *Xenopus* for comparison. For comparisons of tissue dynamics among species, they propose rescaling tissue deformation and synchronizing developmental clocks. They conclude that chicks and *xenopus* share A-P axis asymmetry, as well as the pattern of cell division contributing to elongation of the limb bud.

The study is a valuable contribution to the limb development in describing details of shape change and dynamics over ontogeny and attempting to devise a method for comparisons that helps to understand underlying principles.

My main comments here are about this study's comparison of species. The study lacks a necessary discussion of anuran limb development as qualitatively different than amniotes (chick), first because anuran limb development derives from different tissues and by a different initiating process. The elongation of the frog limb may indeed independently recapitulate the amniote limb developmental characteristics, but the fundamental developmental difference of having the thyroid-dependent metamorphosis in frogs drive limb

initiation, rather than limb buds derived from the early axis/embryonic lateral mesoderm, is not mentioned.

Secondly, the authors conclude - line 403 - "the spatiotemporal expression patterns of hox genes essential for limb development were also found to be conserved"

I wonder about this, as the anuran hind limb zeugopod to autopod boundary is complicated by the proximal tarsal modification into an additional hindlimb segment; along with this morphology, the HoxA11- HoxA13 boundary distinguishing the zeugopod is modified. See: Blanco, M. J., Misof, B. Y., Wagner, G. P., Blanco, M. J., Misof, B. Y., & Wagner, G. P. (1998). Heterochronic differences of Hoxa-11 expression in *Xenopus* fore-and hind limb development: evidence for lower limb identity of the anuran ankle bones. *Development genes and evolution*, 208(4), 175.

Thirdly, the 'paddle' formation in the autopod of anurans at digit formation is also qualitatively different from the amniotes.

from: Fabrezi, M, Goldberg, J, Pereyra, MC. 2017. Morphological variation in anuran limbs: Constraints and novelties. *J. Exp. Zool. (Mol. Dev. Evol.)* 328B: 546– 574.

"Cameron and Fallon (1977) noted how the patterns of digit formation of amphibians differed from that of amniotes. In amphibians, there are no interdigital zones of massive cell death during digit formation. Rather the digits appear to arise by differential proliferation of interdigital and digital cells. Each digital primordium first grows and enlarges, and then it segments to form the interphalangeal joints and the precise number of phalanges (Sanz-Ezquerro and Tickle, 2003). In amphibians and lizards, this process seems to be different from mammals and birds because, even when phalanges differentiate by segmentation, there is no continuous primordium dividing up into as many phalanges. Instead, phalanges appear as cartilaginous condensations that grow in size and then segment to form the next phalange in a proximo-distal direction (Fig. 6)."

All these features would affect the shape dynamics described.

Given that the purpose of the paper is to provide a method for comparing species differences, the acknowledgement of the fundamental differences of highly derived forms requires a review of evidence for homology as a starting point. The differences in gross developmental contexts shouldn't be ignored.

That said, I think this study could potentially provide evidence for similar tissue dynamics *in spite of* the big differences between anurans and chicks, and this discussion is completely missing. For example, if a mouse and chick were compared, there is a strongly supported developmental homology that is 'tinkered' with to create the differences among species, so that the tissue dynamics synchronization is more easily interpretable for similarity and difference from an evolutionary perspective. However, if they find similarities between amniote and anuran limb development, it possibly says more about a deeper question of how evolution is constrained by tissue dynamics even in the case where the 'limb program' is redeployed in a different context (i.e., the post-larval metamorphosis). I would think this is especially important for the stated goal of finding "archetype" forms for organs. (I think they are really assessing is an archetypal ontogeny, not just the resulting form..?)

What an archetype is begs some other questions, which would be worthwhile to present

here. Why are they similar? Any tissue dynamics-based archetype description would need additional discussion of biophysics as a mechanistic contribution adding to this primarily morphological comparison. It could strengthen the discussion.

The conclusion that the pattern of cell division (“cell flow”) contributing to elongation of the limb bud is an important point overall as it contradicts the paradigm that the distal cells are where elongation primarily arises.

Also see

Young, J. J., & Tabin, C. J. (2017). Saunders's framework for understanding limb development as a platform for investigating limb evolution. *Developmental biology*, 429(2), 401-408.

Both D'arcy Thompson's and this study are missing the third dimension of shape, which is critical in cartilage condensation initiation and digit development.

Reviewer #3 (Remarks to the Author):

This manuscript presents a timely exploration of a significant research question, namely, how to establish a standard that facilitates quantitative comparisons of tissue dynamics in homologous organs across different species. The strength of this study lies in its proposal of an approach to map morphogenetic dynamics from one organ to another using a space-time coordinate system. These achievements are made possible through the integration of cutting-edge measurement technologies, advanced data analysis techniques, and a solid foundation in mathematical knowledge. The discovery of the conservation of rescaled tissue dynamics in developing limb buds between tadpoles and chicks may not be considered ground-breaking in itself. However, it is crucial to acknowledge the significance of this finding in light of the absence of rigorous scientific evidence until now, primarily due to the lack of the proposed approach utilized in this study.

The presented results are compelling and well-aligned with the scope of this study. The limitations are adequately described, addressing the essential aspects. While I believe the manuscript is generally of high quality, I have a few suggestions for minor revisions that will further improve the clarity and impact of their work. These suggestions should not require additional experiments or major modifications. While additional experiments to test the robustness and extensibility of this approach under perturbations such as small molecule inhibitors or low temperatures could be envisioned, it is understandable that including them may delay the timely publication, which should be avoided. Expanding the Discussion section, as suggested below, will enhance the overall significance of the results. I believe that incorporating these revisions will significantly enhance the manuscript and make it even more impactful.

1) While the discussion addresses the interpretation of the results, it is recommended to expand on the potential implications of the methods and findings for future research or practical applications in the field. For instance, the authors briefly mention organoid maturation as a target topic for applicability on line 416, but a more detailed explanation would provide greater clarity to the readers. Suggesting potential applications in more depth

would be advantageous for a wide range of readers. Additionally, including a paragraph that highlights the limitations of the study would further enhance its practical implications.

2) There are several suggestions to enhance the clarity and structure of the manuscript. Some statements should be revised to ensure they are within the proper scope. For example, it is important to state in the Abstract that their approach is applicable only a phase when the tissue morphogenesis exhibits simple elongation in the limb bud, as described in the Results section. This will provide readers with a clear understanding of the limitations of the study, as mentioned in my comment #1. Another suggestion is to consider shifting the last section in the Results to the Discussion. This section appears to be more of the authors' views and hypotheses, and may be better suited for the Discussion section rather than being presented as a part of rigid results.

3) Figure 4B: The legend states that the white points indicate the closest chick stage to each Xenopus stage, and the black points indicate the Xenopus to chick stage. However, in Figure 4B, the representation of white and black points seems to be opposite to the description in the legend. Please verify the correctness.

4) Figure 4E: The authors mentioned that the choice of initial times for each target affects the level of similarity, but they only show the case when the Xenopus stage is fixed. It would be beneficial to present the results in a round-robin manner, considering different initial times for each target. Also, it would be helpful to provide a brief explanation or interpretation as to why the difference level is smaller at chick stage 21 compared to earlier or later stages. The authors should describe what the error bars represent and provide the number of samples used in the analysis in the legend.

5) Figure 5: The authors claim a high degree of similarity in ξ -space (line 353), which is an important result. It would be beneficial to quantify the level of similarity to provide a more quantitative assessment of the observed similarity.

6) Materials and Methods, line 441: "All animal handling was performed under appropriate anesthesia ..". It is recommended to provide a detailed description of the anesthesia protocol used in the study for the purpose of reproducibility.

**Replies to reviewers' comments**

First, we would like to thank all the reviewers for their very valuable comments and questions. We
have listed our responses to the individual comments below.

6 **Replies to Reviewer #1's Comments**

*Reviewer #1 (Remarks to the Author):*

*This is an interesting study attempting to find a common morphometric plan underlying the*
*development of hindlimbs in two vertebrate model organisms which differ in size, shape and*
*developmental timing. It combines mathematical modelling with cell tracking in vivo, and builds on a*
*previous study of the chicken hind limb bud. The study uses a very clever method of labelling many*
*small groups of cells in the mesenchyme along with mathematical modelling which supports an*
*“archetype” for stage 52 to 54. This stage interval is about 5 days of Xenopus tadpole development*
*(starting 9 days after the buds appear, and finishing as digits begin to condense). At these stages the*
*mesenchyme anterior-posterior axis is well established, and cells fated to form the stylopod, zeugopod*
*and autopod can be identified (Tschumi 1957). Effectively, the study and model covers the period when*
*zeugopodal cartilage elements condense and when digits and interdigits are being defined in the*
*autopod. The latter has been recently shown to be recapitulated in in vitro culture in mice, indicating*
*self-organising ability of the limb bud mesenchyme (Fuiten et al PMID: 36994104)The model is*
*backed up with some nice RNAscope data of anterior posterior marker hox genes. There are also of*
*course more interdigital regions in frog hindlimb (5 digits) vs. chick hindlimb (4 digits) and I can't see*
*this reflected in the model or discussed.*

*Major: The paper would benefit from some more background into the model to be more accessible to*
*biologist readers. The discussion is brief and includes no citations, so is more a conclusion. I think at*
*the least it would be good to address the limitations of this model in terms of the stages it covers, and*
*perhaps to expand the discussion to include reference to the work of others – e.g. What is known about*
*cell division rate and orientation in these models, or in other vertebrates, and how would this direct*
*morphogenetic growth in a comparable way? Additionally I suggest changes that I think would assist*
*with broad readership appeal as “minor” below.*

32 **Reply to Reviewer #1's major comments:**

Thank you very much for your valuable comments. First, please let us clarify the usage of the term
“model”. In this study, we used a Bayesian statistical model to reconstruct smooth tissue deformation
maps from noisy cell trajectory data, where the term “deformation map” means the positional
correspondence of each point within a tissue at different time points or the trajectory of each point by

which deformation characteristics can be calculated. Thus, the spatiotemporal patterns of quantified
local tissue deformation characteristics (heatmaps in Fig. 2) and the cell flows under the ξ coordinate
system (Fig. 3) are real values, not virtual (model) values. In this sense, the difference in the number
of digits between species does not need to be accounted for *a priori* to calculate these quantities. In
addition, as stated in the original manuscript,

“Anatomically, the interspecies differences in the relative lengths of the tarsals and zeugopods and the
number of formed digits begin to appear around $t_{Xenopus}=53.5$ (*xenopus*) and $t_{Chick}=27$ (chick). Thus,
these highly consistent cell trajectories in the ξ -space suggest that temporal changes in the relative
positions of cells within the developing tissues are well conserved regardless of the state of cartilage
differentiation.”

After the basic skeletal patterns are formed, species-specific tissue deformation begins to appear.
Regarding this point, we have added the following sentence in the Results section of the revised
manuscript:

“For example, regarding cell/tissue behavior during digit formation within the paddle-like autopod
region, amphibians and amniotes have been reported to have qualitative differences in the rate of cell
death in the interdigital zone and in the segmentation processes of digits (31).”

Furthermore, the following text regarding the quantified spatiotemporal patterns of tissue deformation
characteristics has been added to the Results section:

Regarding A-P asymmetric growth, “Further, in our previous study on chick limb development, we
showed that this A-P asymmetric area growth rate at the tissue level is quantitatively consistent with
the positional dependence of the cell cycle time (14).” Regarding the almost homogeneous elongation
rate along the P-D axis, “This fact indicates that, similar to the chick case (14), the P-D elongation of
a *Xenopus* limb bud cannot be explained by the classical model that limb bud elongation is caused
primarily by proliferation of distal cells (25-27, and see also (24) that nicely reviews the history of
“proliferation gradient” model); it should be noted that the factors that drive this anisotropic local
tissue deformation remain unknown.”

We have substantially revised the Discussion section in accordance with the three reviewers’
comments. The major modifications are as follows:

(i) We have added the following sentences describing the limitations of our methodology (please see
the subsection “Representation of space-time information and limitations”):

“With respect to time (τ), we propose a method for achieving synchronization of developmental time
between species based on geometrical information, i.e., cell flow under the ξ -coordinate system. The
method has a limitation that it functions effectively only when the transformation of time coordinates
between species adheres to the conditions of being both bijective and continuous. These conditions

can be compromised when significant interspecies differences in the flow arise; in the case of chick
and *Xenopus* limb development, (geometrical) synchronization was not possible after the phase when
basic digit patterning was established. We should note that this does not necessarily imply only a
negative aspect, because the method conversely could also serve to detect the emergence of
interspecies differences in tissue dynamics, as shown above.”

(ii) We have also modified the text regarding future challenges:

“In this study, we decomposed a tissue deformation map as the product of the average growth $\mathbf{L}(t)$
(i.e., species-specific tissue size and aspect ratio at time t) and the rescaled tissue dynamics $\xi(\xi_0, \tau)$
(Eq. 1) and focused on similarities in the latter. However, how the former, species-specific tissue size,
is determined remains a critical issue because it not only satisfies pure scientific interest but also is
closely linked to applied research. For example, current organoids can reproduce the differentiated
states of individual organs, but remain considerably distant from being fully functional, mature organs
of adequate size (37). Identifying the factors that determine interspecies differences in the size of
homologous organs will introduce the possibility of regulating the size of immature organoids. A group
of genes associated with species differences whose expression patterns are not conserved in the τ - ξ
coordinate system will provide clues to elucidate the molecular mechanisms responsible for organ size
determination.”

(iii) The following three items have also been added to the Discussion section in response to
suggestions from other reviewers.

“Spatially, positions in the rescaled space (ξ), rather than those on an absolute scale, would be encoded
as gene expression vectors in a manner common to different species. Changes in tissue size (or more
precisely, average deformation $\mathbf{L}(\tau)$) are cancelled out in ξ -space, which allows direct comparison of
spatial representations within a tissue at different time points. This perspective will be crucial for
advancing the previous theoretical frameworks on positional-information coding in static fields (33,
34) to encompass dynamic scenarios.”

“As described in the Introduction, the trigger/timing of limb development are very different between
chick and *Xenopus* (Keenan and Beck, Dev. Dyn., 2016). Despite such qualitative differences, it is
surprising that tissue dynamics are well conserved, which suggests that the evolution of limb
morphogenesis is constrained by common physical processes as well as conserved signaling pathways
and gene expression patterns across species. Clearly, such an interdisciplinary understanding is
necessary to determine what an archetype of tissue dynamics is (including knowing if an archetype
itself really exists). In the context of limb development, clarifying the physicochemical factors
responsible for the nearly uniform elongation rate along the P-D axis and asymmetric

growth/deformation along the A-P axis, as revealed in our analysis (Figs. 3C-H), will be an important
clue to understanding the constraints.”

“Especially, since three-dimensional (3D) features of organ morphology become more pronounced in
the later stages of development, 3D map reconstruction would be critical. In the context of limb
development, at stages following the establishment of the skeletal patterning (e.g., after St. 30 in the
chick case), the shapes of all digits become more distinct, while simultaneously assuming a more 3D
arrangement relative to each other. Thus, extending the analysis to a 3D analysis will be essential for
a deeper understanding of species differences in tissue dynamics.”

Minor:

*Throughout: species names in italics*

**Reply:** We have corrected all mentions of species names so that they are in italics.

*Abstract: include full species names for both chicken and Xenopus*

**Reply:** We have added this information in the revised manuscript.

*line 71 I am not familiar with the exact meaning of “tissue deformation” in this context, and think it*
*would be more accessible to wider readership if this was briefly explained.*

**Reply:** Thank you for your comment. We have added a brief explanation of tissue deformation as
follows:

(original)

... the tissue deformation dynamics during animal development (6).

(revised)

... the tissue deformation dynamics during animal development that include spatio-temporal patterns
of area/volume change of each local tissue piece and the extent/direction of its stretching or shrinking
(6).

*Line 84: similarly, what is a tissue deformation map?*

**Reply:** We have added a brief explanation of a tissue deformation map as follows:

(original)

... to obtain quantitative tissue deformation maps for the developmental processes of organs...

(revised)

... to obtain quantitative tissue deformation maps (i.e., the positional correspondence of each point
within a tissue at different time points or the trajectory of each point by which deformation

characteristics can be calculated) for the developmental processes of organs...

*Line 114: can you explain why “on the dorsal ventral boundary plane” and what this means? Is the*
*grid focused somehow so that epithelial cells are not triggered to express GFP? The image in figure*
*2A (and S1) is really nice but I cannot tell if it is mesenchymal only – were two lasers used to focus on*
*a single spot, and only where they meet the heat shock promoter activates?*

**Reply:** Thank you for this comment. For convenience, here we were referring to the frontal section at
the mid dorsoventral (D-V) level where the cross-sectional area of a limb bud becomes maximal as
the “D-V boundary plane”. To avoid any confusion, we have decided not to use the term “dorsal-
ventral boundary” in the revised manuscript. The focus on that plane is because the distinctive event
of cartilage formation during limb development occurs around the D-V boundary, not at the
dorsal/ventral end. Regarding GFP induction, laser irradiation can be focused to a certain D-V level to
some extent, but each spot can become elongated in the D-V direction. However, we found that the
shape of each spot was well maintained during our measurement time interval (around 24 hours), and
that even if we changed the focus of the microscope in the D-V direction, the spot position hardly
changed in the plane spanned by the A-P and P-D axes. This means that the D-V dependence of the
in-plane deformation is sufficiently small, at least in the measurement interval. As for this point, we
have added the explanation in the Materials and Methods as follows:

“Since our aim was to compare the 2D deformation dynamics on the frontal plane at the mid D-V level
with the maximal area of a limb bud in *Xenopus* with that previously reported for chick (14), we
focused the irradiation within this plane (Fig. 2A). The heat-shock treatment was performed as
described in our previous work (20). The diameter of a single labeled spot on the plane was typically
20-30 μm , a size equivalent to a few cells. Although laser irradiation can be focused to a certain D-V
level to some extent, each spot could become elongated in the D-V direction. However, we found that
the in-plane shape of each spot was well maintained during our measurement time interval (around 24
171 hours), and that even if we changed the focus of the microscope in the D-V direction, the spot position
hardly changed in the plane. This means that the D-V dependence of the in-plane deformation is
sufficiently small, at least in the measurement interval.”

*Line 120/121 state the Nieuwkoop and Faber stages used?*

**Reply:** The staging method we adopted is described in the statement immediately following. In a
previous study, we proposed an objective staging method based on the contour shape of a limb bud,
and we adopted that method here. This was because the developmental rate of *Xenopus*, a cold-blooded
animal, varies significantly, and there is not always a precise alignment between stage values and

actual time. This point was stated in the Materials and Methods:
“Staging of each individual was based on a previously proposed morphometric staging method (21);
briefly, digitized outlines of the limb buds were approximated using elliptic Fourier descriptors, and a
continuous stage value, not discrete values as in traditional staging (e.g., 51 and 52), was assigned to
each individual based on its coefficients. In this study, this morphometric stage was denoted by
$t_{Xenopus}$.”

*Line 135: Xenopus limb bud stages are points on a continuum, but how did you accurately subdivide*
*each stage in to 10th stages?*

**Reply:** As stated above, *Xenopus* staging was based on a previously proposed morphometric method.
Briefly, digitized outlines of the limb buds were approximated using elliptic Fourier descriptors, and
a continuous stage value, not discrete values as in traditional staging (e.g., 51 and 52), was assigned
to each sample based on its coefficients. For each individual, the positional coordinates of the
fluorescently labeled spots were measured at 2 to 4 time points approximately every 24 hours, and the
coordinates at timepoints between the measurements were obtained by linear interpolation. That is, as
shown in Fig. S1C, for each sample, we obtained cell trajectory data for a time window represented
as a line segment on the number line. Samples containing cell trajectory data within each subdivided
interval were used to reconstruct the deformation dynamics during that interval. This point is explained
in the Materials and Methods as follows:

“Staging of each individual was based on a previously proposed morphometric staging method (21);
briefly, digitized outlines of the limb buds were approximated using elliptic Fourier descriptors, and a
continuous stage value, not discrete values as in traditional staging (e.g., 51 and 52), was assigned to
each individual based on its coefficients. In this study, this morphometric stage was denoted by $t_{Xenopus}$.
Each limb bud was resized to a previously determined typical value corresponding to its morphometric
stage. The spatial coordinates of the fluorescently labeled spots were measured at 2 to 4 time points
approximately every 24 hours for each individual, and the coordinates at the timepoints between the
measurements were obtained by linear interpolation. The period from $t_{Xenopus} = 50.6$ to $t_{Xenopus} = 54.4$
was divided into nine roughly equally spaced time intervals (each interval corresponded to a 0.4-0.5
stage increment), and data from the individuals included in each interval were integrated to reconstruct
the tissue deformation map for that interval.”

*Line 136 state HH stages*

**Reply:** Yes, we based our chick staging on the Hamburger-Hamilton table. A stage value was assigned
to each sample using increments of 0.5 (not necessarily integers) based on incubation time. In our

previous study, we quantified the deformation map for every 12-hour interval. Thus, each heatmap
shown in Fig. 2E is the result of each 12-hour interval. Since the developmental rate of chickens,
which are warm-blooded animals, closely matched real time, we staged them based on real time rather
than morphology. This point is stated in the Materials and Methods as follows:
“For chick hindlimb development, we used a previously reported tissue deformation map
$\mathbf{x} = \phi_{\text{Chick}}(\mathbf{X}, t)$ (14). The only minor modification from the previous study was that we changed the
staging method. In the previous study, a tissue deformation map was quantified for every 12-hour
interval, and for each time point, we assigned the integer value of the Hamburger-Hamilton stage with
a shape closest to that observed at the time point. In the present study, staging was performed using
increments of 0.5 (not necessarily integers) based on incubation time.”

*Line 139 citations are needed to support this “Xenopus limb buds include the prospective autopod*
*(toe-to-ankle), zeugopod (lower leg), and stylopod (upper leg) regions. In contrast, chick limb buds*
*contain mainly the former two, while the stylopod is embedded in the trunk.*

**Reply:** We modified the corresponding sentences as follows: “*Xenopus* limb buds include the
prospective autopod (toe-to-ankle), zeugopod (lower leg), and stylopod (upper leg) regions (22). In
contrast, as shown later, based on the inverse mapping of cartilage patterns, chick limb buds contain
mainly the former two, while the stylopod is embedded in the trunk.”

*Line 173 gene names in italics*

**Reply:** We have made this correction.

*Line 229 “we can intuitively see...” please explain what we are supposed to be seeing because I cannot*
*make this conclusion from the coloured vectors in figure 3C and D. Are there even interdigits present*
*at stage 52? Maybe this refers to another figure?*

**Reply:** Thank you for this comment. The colored vectors represent the movement of the limb bud
mesenchymal cells when observed in the ξ coordinate system. What we wanted to state was that, in
the internal tissues of a limb bud, both species share a similar flow oriented from the posterior to the
anterior side, while the direction of the arrow around the limb bud boundaries (e.g., anterior boundary)
is not necessarily similar. We have modified the text as follows:

(original)

“we can intuitively see that the cell trajectories of the internal tissues that will form future skeletal
structures are more consistent between species than those near the tissue boundaries (Fig. 3C, **D**)”

(revised)

“we can intuitively see that the cell trajectories (the flow patterns in the ξ coordinate system) of the
internal tissues that will form future skeletal structures are more consistent between species (i.e.,
basically oriented from posterior to anterior) than those near the tissue boundaries (Fig. 3C, D)”

*Paragraph starting line 236: what criteria did you use/match to set the clock for each organism (e.g*
*cartilage condensation, gene expression?)*

**Reply:** Thank you for this comment. We introduced the concept of a common clock, which is
represented as an abstract one-dimensional curve, and we regarded the developmental stages of each
species as time coordinates of the common clock. Then, determining the correspondence of the
developmental stages between species (i.e., synchronizing the developmental times) means giving the
coordinate transformations between those time coordinates. Here, we devised a method of
synchronization (or coordinate transformation) based solely on the geometric information, i.e., tissue
deformation dynamics, not on changes in cellular states such as gene expression or cartilage
condensation. As shown in Fig. 3, we defined a spatial coordinate system, ξ , in which tissue
deformation is represented as a cell flow. We synchronized the developmental time by adjusting the
scale interval of the time axis of both species so that the difference between cell trajectories starting
from the same initial position was as small as possible. The introduction of mathematical concepts is
inevitable for a more precise definition of synchronization. The main text is limited to an intuitive
explanation, while the Materials and Methods text provides a more rigorous explanation (please see
the subsection “Common clock T and synchronization between species”). Regarding this point, the
main text states the following:

“As a way to find the composite map that relates stages between chick and *Xenopus* (denoted by

$\Psi_C \circ \Psi_X^{-1}$ and $\Psi_X \circ \Psi_C^{-1}$), we aligned the scales of their temporal axes such that the mean difference

in cell trajectories from the same initial position in the ξ -space is minimized (see Materials and
Methods for details).”

*Line 487 H2O2*

**Reply:** We have rewritten this correctly as H₂O₂.

*Line 499 I'm still not understanding what you mean by dorsal ventral boundary: the mesenchyme does*
*not really have such a boundary (unlike the epithelial AER?) – does this mean cells were labelled in*
*the centre relative to the DV axis, where lmx1b positive and negative cells meet? Could you perhaps*
*illustrate this better with a figure?*

**Reply:** As explained above, here we were referring to the frontal section at the mid dorsoventral (D-
289 V) level where the cross-sectional area of a limb bud becomes maximal as the “D-V boundary plane”.
To avoid any confusion, we have decided not to use the term “dorsal- ventral boundary” in the revised
manuscript. We have modified Fig. S1 to illustrate this.

*Line 669 Gene names in italics*

**Reply:** We have made this correction.

*Figure 1: I find this legend to be a bit overlong, it should really be re-written to just describe what is*
*shown and the rest can be incorporated into the text? In particular, having a study aim in the legend*
*seems out of place.*

**Reply:** Thank you very much for this comment. We have revised the legend of Figure 1 accordingly.
As you pointed out, we agree that including the study aim is out of place; thus, we deleted it. We then
confirmed that the rest of the text corresponds to the figure.

*Figure 2 The Xenopus A-P axis is posturally inverted: shh marking the ZPA clearly shows this for the*
*stages of this study: Endo et al 1997 PMID: 9186057, Keenan and Beck 2016 PMID: 26404044, Wang*
*et al 2015 PMID: 26527308. I can see you’ve inverted the limbs in this figure to account for this but*
*it should be stated.*

**Reply:** Thank you for this comment. At your suggestion, we have stated the following in the legend
of Fig. 2: “Note that the images of the *Xenopus* limb bud, except for the top-left photo in panel (A),
are inverted in the A-P direction to have the posterior side facing downward.”

*I’m not clear on how the maps are made- 1D or is information captured from multiple layers and*
*stacked? If the former then how do you account for cells that move or divide dorsoventrally in the*
*mesenchyme?*

**Reply:** As stated above, we are focusing on the 2D deformation dynamics of mesenchymal tissue on
the frontal plane at around the mid D-V level where the cross-sectional area of a limb bud becomes
maximal. In addition, even when each spot became elongated in the D-V direction, we found that the
shape of each spot was well maintained during our measurement time interval (around 24 hours), and
that when we changed the focus of the microscope in the D-V direction, the spot position hardly
changed in the plane. This means that the D-V dependence of the in-plane deformation is sufficiently
small, at least in the measurement interval. Another important piece of information is that tissue-level

deformation is not calculated from the movement of cells within each spot, but from the change in the
relative positions among the labeled spots. Therefore, if the relative positional changes between spots
over the entire limb bud are reproducible, the deformation dynamics can be correctly reconstructed.
In that sense, our data are highly reproducible among samples, and the resultant reconstructed tissue
deformation dynamics are reliable. Regarding the latter point, we have added the following sentences
in the Materials and Methods:

“It should be noted that tissue-level deformation is not calculated from the movement of cells within
each spot, but from the change in the relative positions among the labeled spots. Therefore, if the
relative positional changes between spots over the entire limb bud are reproducible, the deformation
dynamics can be correctly reconstructed. In that sense, our data are highly reproducible among
samples, and the resultant reconstructed tissue deformation dynamics are reliable.”

*Figure 3 rather than using left and right, fro C and D maybe label all the panels ? In the legend, the*
*final statement “Note that the dynamics for the prospective autopod and zeugopod are compared” at*
*the end does not seem to obviously relate to anything shown in the figure?*

**Reply:** According to this comment, we relabeled Fig. 3 (please see Fig. 3 and its legend in the revised
manuscript). The final statement, “Note that the dynamics for the prospective autopod and zeugopod
are compared” is related to the figure because the flow pattern in the ξ coordinate system changes
depending on the region to be analyzed within the limb bud (as analyzed in Fig. 4H). We have slightly
modified that sentence as follows:

(original)

“Note that the dynamics for the prospective autopod and zeugopod are compared.”

(revised)

“The cell flows in the ξ coordinate system shown in panels (C) and (D) correspond to the
morphogenesis of the region consisting of prospective autopods and zeugopods.”

**Replies to Reviewer #2's Comments**

*Reviewer #2 (Remarks to the Author):*

*ms review*

*NCOMMS-23-19472-T*

*The goal of the paper is to explore whether there are conserved 'archetypal' growth dynamics in organs*
*detectable among species. The authors produce tissue deformation maps through developmental*
*stages of chick and Xenopus for comparison. For comparisons of tissue dynamics among species, they*
*propose rescaling tissue deformation and synchronizing developmental clocks. They conclude that*
*chicks and xenopus share A-P axis asymmetry, as well as the pattern of cell division contributing to*
*elongation of the limb bud. The study is a valuable contribution to the limb development in describing*
*details of shape change and dynamics over ontogeny and attempting to devise a method for*
*comparisons that helps to understand underlying principles.*

*My main comments here are about this study's comparison of species. The study lacks a necessary*
*discussion of anuran limb development as qualitatively different than amniotes (chick), first because*
*anuran limb development derives from different tissues and by a different initiating process. The*
*elongation of the frog limb may indeed independently recapitulate the amniote limb developmental*
*characteristics, but the fundamental developmental difference of having the thyroid-dependent*
*metamorphosis in frogs drive limb initiation, rather than limb buds derived from the early*
*axis/embryonic lateral mesoderm, is not mentioned.*

**Reply:** First, we would like to sincerely thank you for your very valuable comments. We completely
agree that stating the differences in gross developmental contexts is biologically very important in
comparing chickens and frogs. We have modified the manuscript to reflect these comments as much
as possible.

According to this comment, we have added the following sentences in the Introduction and Discussion
sections:

(Introduction)

"Both species share many developmental characteristics, including major signaling and gene
expression patterns, whereas the trigger and timing of development are clearly different. Limb buds
of amniotes including chick develop concurrently with the main body axis formation of an embryo
and arise from the lateral plate mesoderm. In contrast, limb development in *Xenopus* proceeds as one
of the thyroxine (thyroid hormone)-dependent events in metamorphosis after embryonic stage (17),
and the precise origin of a limb bud is difficult to determine (18). Through the comparison of

homologous organs with such qualitative differences, we inquired into the existence of archetypal
tissue dynamics.”

(Discussion)

“As described in the Introduction, the trigger/timing of limb development are different between chick
and *Xenopus*. Despite such qualitative differences, it is surprising that tissue dynamics are well
conserved, which suggests that the evolution of limb morphogenesis is constrained by common
physical processes, as well as conserved signaling pathways and gene expression patterns across
species. Clearly, such an interdisciplinary understanding is necessary to determine what an archetype
of tissue dynamics is (including knowing if an archetype itself really exists). In the context of limb
development, clarifying the physicochemical factors responsible for the nearly uniform elongation rate
along the P-D axis and asymmetric growth/deformation along the A-P axis, as revealed in our analysis
(Figs. 3C-H), will be an important clue to understanding the constraints.”

*Secondly, the authors conclude - line 403 - “the spatiotemporal expression patterns of hox genes*
*essential for limb development were also found to be conserved”*

*I wonder about this, as the anuran hind limb zeugopod to autopod boundary is complicated by the*
*proximal tarsal modification into an additional hindlimb segment; along with this morphology, the*
*HoxA11- HoxA13 boundary distinguishing the zeugopod is modified. See:*

*Blanco, M. J., Misof, B. Y., Wagner, G. P., Blanco, M. J., Misof, B. Y., & Wagner, G. P. (1998).*
*Heterochronic differences of Hoxa-11 expression in Xenopus fore-and hind limb development:*
*evidence for lower limb identity of the anuran ankle bones. Development genes and evolution, 208(4),*
*175.*

**Reply:** Thank you for this comment. We read the above paper, and we now have a better understanding
of the differences between forelimbs and hindlimbs. In this paper, the expression of *HoxA11* in the
hindlimb was examined by RNAscope, and we found that the obtained pattern was somewhat different
from that examined by ordinary *in situ* hybridization in the Blanco 1998 paper. For example, in the
Blanco 1998 paper, the expression in later stages was very weak, whereas our results show a more
regional pattern with a broad, strong signal in the zeugopod. We think that this difference is due to
detection sensitivity. Regarding the comparison of expression patterns in chick and *Xenopus* hindlimbs,
as shown in the right panel of Fig. 5B, the range of *Hoxa11* expression in ξ -space was very similar in
both species, at least when basic limb skeletal patterning was done. In contrast, the somewhat
ambiguous overlapping of the *Hoxa11* expression ranges between species in the early stages is
probably because those stages correspond to the transient period in which the distal end of the *Hoxa11*
expression region shifts from the tip of limb bud to the more proximal side.

We have added the following sentences in the Results.

“It should be noted that the overlap of *Hoxa11* expression ranges of both species was somewhat
ambiguous in the early stages, which is probably because the stages correspond to the transient period
in which the distal end of the *Hoxa11* expression region shifts from the tip of the limb bud to the more
proximal side.”

Lastly, the differences between the forelimb and the hindlimb described in the Blanco 1998 paper are
so important that we will take them into consideration when studying the similarities/differences in
those tissue dynamics in future.

*Thirdly, the ‘paddle’ formation in the autopod of anurans at digit formation is also qualitatively*
*different from the amniotes.*

*from: Fabrezi, M, Goldberg, J, Pereyra, MC. 2017. Morphological variation in anuran limbs:*
*Constraints and novelties. J. Exp. Zool. (Mol. Dev. Evol.) 328B: 546– 574.*

*“Cameron and Fallon (1977) noted how the patterns of digit formation of amphibians differed from*
*that of amniotes. In amphibians, there are no interdigital zones of massive cell death during digit*
*formation. Rather the digits appear to arise by differential proliferation of interdigital and digital cells.*
*Each digital primordium first grows and enlarges, and then it segments to form the interphalangeal*
*joints and the precise number of phalanges (Sanz-Ezquerro and Tickle, 2003). In amphibians and*
*lizards, this process seems to be different from mammals and birds because, even when phalanges*
*differentiate by segmentation, there is no continuous primordium dividing up into as many phalanges.*
*Instead, phalanges appear as cartilaginous condensations that grow in size and then segment to form*
*the next phalange in a proximo-distal direction (Fig. 6).”*

*All these features would affect the shape dynamics described. Given that the purpose of the paper is*
*to provide a method for comparing species differences, the acknowledgement of the fundamental*
*differences of highly derived forms requires a review of evidence for homology as a starting point. The*
*differences in gross developmental contexts shouldn’t be ignored.*

**Reply:** We have added the following sentence in the Results section:

“For example, regarding cell/tissue behavior during digit formation within the paddle-like autopod
region, amphibians and amniotes have been reported to have qualitative differences in the rate of cell
death in the interdigital zone and in the segmentation processes of digits(31).”

*That said, I think this study could potentially provide evidence for similar tissue dynamics *in spite*
*of* the big differences between anurans and chicks, and this discussion is completely missing. For*
*example, if a mouse and chick were compared, there is a strongly supported developmental homology*
*that is ‘tinkered’ with to create the differences among species, so that the tissue dynamics*

*synchronization is more easily interpretable for similarity and difference from an evolutionary*
*perspective. However, if they find similarities between amniote and anuran limb development, it*
*possibly says more about a deeper question of how evolution is constrained by tissue dynamics even*
*in the case where the ‘limb program’ is redeployed in a different context (i.e., the post-larval*
*metamorphosis). I would think this is especially important for the stated goal of finding “archetype”*
*forms for organs. (I think they are really assessing is an archetypal ontogeny, not just the resulting*
*form...?)*

*What an archetype is begs some other questions, which would be worthwhile to present here. Why are*
*they similar? Any tissue dynamics-based archetype description would need additional discussion of*
*biophysics as a mechanistic contribution adding to this primarily morphological comparison. It could*
*strengthen the discussion.*

**Reply:** We appreciate this valuable comment. As described above, according to the comments received,
we have added the following sentences in the Discussion section:

“As described in the Introduction, the trigger/timing of limb development are very different between
chick and *Xenopus*. Despite such qualitative differences, it is surprising that tissue dynamics are well
conserved, which suggests that the evolution of limb morphogenesis is constrained by common
physical processes, as well as conserved signaling pathways and gene expression patterns across
species. Clearly, such an interdisciplinary understanding is necessary to determine what an archetype
of tissue dynamics is (including knowing if an archetype itself really exists). In the context of limb
development, clarifying the physicochemical factors responsible for the nearly uniform elongation rate
along the P-D axis and asymmetric growth/deformation along the A-P axis, as revealed in our analysis
(Figs. 3C-H), will be an important clue to understanding the constraints.”

*The conclusion that the pattern of cell division (“cell flow”) contributing to elongation of the limb*
*bud is an important point overall as it contradicts the paradigm that the distal cells are where*
*elongation primarily arises.*

*Also see*

*Young, J. J., & Tabin, C. J. (2017). Saunders's framework for understanding limb development as a*
*platform for investigating limb evolution. *Developmental biology*, 429(2), 401-408.*

**Reply:** We have added the following sentence:

“This fact indicates that, similar to the chick case (14), the P-D elongation of a *Xenopus* limb bud
cannot be explained by the classical model that limb bud elongation is caused primarily by
proliferation of distal cells (25-27, and see also (24) that nicely reviews the history of “proliferation
gradient” model)”

*Both D'arcy Thompson's and this study are missing the third dimension of shape, which is critical in*
*cartilage condensation initiation and digit development.*

**Reply:** We agree with this comment. In this study, we analyzed the two-dimensional tissue
deformation dynamics of mesenchyme in the frontal plane at around the middle of the D-V axis, where
basic skeletal patterning occurs. At stages following the establishment of the skeletal patterning (e.g.,
after St. 30 in the chick case), the shapes of all the digits become more distinct, while simultaneously
assuming a more three-dimensional arrangement relative to each other. Species differences in the
morphology and the proportion of anatomical structures will be stronger in later stages, and thus
extending the analysis to a 3D analysis will be essential for a deeper understanding of species
differences in tissue dynamics. We have added the following text in the Discussion section:

“Especially, since three-dimensional (3D) features of organ morphology become more pronounced in
the later stages of development, 3D map reconstruction would be critical. In the context of limb
development, at stages following the establishment of the skeletal patterning (e.g., after St. 30 in the
chick case), the shapes of all digits become more distinct, while simultaneously assuming a more 3D
arrangement relative to each other. Thus, extending the analysis to a 3D analysis will be essential for
a deeper understanding of species differences in tissue dynamics.”

**Reply to reviewer #3's comments:**

*Reviewer #3 (Remarks to the Author):*

*This manuscript presents a timely exploration of a significant research question, namely, how to*
*establish a standard that facilitates quantitative comparisons of tissue dynamics in homologous organs*
*across different species. The strength of this study lies in its proposal of an approach to map*
*morphogenetic dynamics from one organ to another using a space-time coordinate system. These*
*achievements are made possible through the integration of cutting-edge measurement technologies,*
*advanced data analysis techniques, and a solid foundation in mathematical knowledge. The discovery*
*of the conservation of rescaled tissue dynamics in developing limb buds between tadpoles and chicks*
*may not be considered ground-breaking in itself. However, it is crucial to acknowledge the significance*
*of this finding in light of the absence of rigorous scientific evidence until now, primarily due to the*
*lack of the proposed approach utilized in this study.*

*The presented results are compelling and well-aligned with the scope of this study. The limitations are*
*adequately described, addressing the essential aspects. While I believe the manuscript is generally of*
*high quality, I have a few suggestions for minor revisions that will further improve the clarity and*
*impact of their work. These suggestions should not require additional experiments or major*
*modifications. While additional experiments to test the robustness and extensibility of this approach*
*under perturbations such as small molecule inhibitors or low temperatures could be envisioned, it is*
*understandable that including them may delay the timely publication, which should be avoided.*
*Expanding the Discussion section, as suggested below, will enhance the overall significance of the*
*results. I believe that incorporating these revisions will significantly enhance the manuscript and make*
*it even more impactful.*

**Reply:** Thank you very much for your positive comments. Below is a list of responses to your
suggestions.

*1) While the discussion addresses the interpretation of the results, it is recommended to expand on the*
*potential implications of the methods and findings for future research or practical applications in the*
*field. For instance, the authors briefly mention organoid maturation as a target topic for applicability*
*on line 416, but a more detailed explanation would provide greater clarity to the readers. Suggesting*
*potential applications in more depth would be advantageous for a wide range of readers. Additionally,*
*including a paragraph that highlights the limitations of the study would further enhance its practical*
*implications.*

**Reply:** The Discussion section has been substantially revised in accordance with the comments

received.

(i) We have added the following sentences in “Future challenges” subsection of the Discussion section:
“In this study, we decomposed a tissue deformation map as the product of the average growth $\mathbf{L}(t)$,
that determines species-specific tissue size and aspect ratio at time t , and the rescaled tissue dynamics
$\xi(\xi_0, \tau)$ (Eq. 1) and focused on similarities in the latter. However, how the former, species-specific
tissue size, is determined remains a critical issue because it not only satisfies pure scientific interest
but also is closely linked to applied research. For example, current organoids can reproduce the
differentiated states of individual organs, but remain considerably distant from being fully functional,
mature organs of adequate sizes (37). Identifying the factors that determine interspecies differences in
the size of homologous organs will introduce the possibility of regulating the size of immature
organoids.”

(ii) We have also added the following sentences describing the limitations of our methodology to the
Discussion section (please see the subsection “Representation of space-time information and
limitations”):
“With respect to time (τ), we propose a method for achieving synchronization of developmental time
between species based on geometrical information, i.e., cell flow under the ξ -coordinate system. The
method has a limitation that it functions effectively only when the transformation of time coordinates
between species adheres to the conditions of being both bijective and continuous. These conditions
can be compromised when significant interspecies differences in the flow arise; in the case of chick
and *Xenopus* limb development, (geometrical) synchronization was not possible after the phase when
basic digit patterning was established. We should note that this does not necessarily imply only a
negative aspect, because the method conversely could also serve to detect the emergence of
interspecies differences in tissue dynamics, as shown above.”

(iii) In addition, the following three items have also been added to the Discussion section in response
to suggestions from other reviewers.
“Spatially, positions in the rescaled space (ξ), rather than those on an absolute scale, would be encoded
as gene expression vectors in a manner common to different species. Changes in tissue size (or more
precisely, average deformation $\mathbf{L}(\tau)$) are cancelled out in ξ -space, which allows direct comparisons of
spatial representations within a tissue at different time points. This perspective will be crucial for
advancing the previous theoretical frameworks on positional-information coding in static fields (33,
34) to encompass dynamic scenarios.”

“As described in the Introduction, the trigger/timing of limb development are very different between
chick and *Xenopus*. Despite such qualitative differences, it is surprising that tissue dynamics are well

conserved, which suggests that the evolution of limb morphogenesis is constrained by common
physical processes, as well as conserved signaling pathways and gene expression patterns across
species. Clearly, such an interdisciplinary understanding is necessary to determine what an archetype
of tissue dynamics is (including knowing if an archetype itself really exists). In the context of limb
development, clarifying the physicochemical factors responsible for the nearly uniform elongation rate
along the P-D axis and asymmetric growth/deformation along the A-P axis, as revealed in our analysis
(Figs. 3C-H), will be an important clue to understanding the constraints.”

“Especially, since three-dimensional (3D) features of organ morphology become more pronounced in
the later stages of development, 3D map reconstruction would be critical. In the context of limb
development, at stages following the establishment of the skeletal patterning (e.g., after St. 30 in the
chick case), the shapes of all the digits become more distinct, while simultaneously assuming a more
3D arrangement relative to each other. Thus, extending the analysis to a 3D analysis will be essential
for a deeper understanding of species differences in tissue dynamics.”

*2) There are several suggestions to enhance the clarity and structure of the manuscript. Some*
*statements should be revised to ensure they are within the proper scope. For example, it is important*
*to state in the Abstract that their approach is applicable only a phase when the tissue morphogenesis*
*exhibits simple elongation in the limb bud, as described in the Results section. This will provide*
*readers with a clear understanding of the limitations of the study, as mentioned in my comment #1.*
*Another suggestion is to consider shifting the last section in the Results to the Discussion. This section*
*appears to be more of the authors' views and hypotheses, and may be better suited for the Discussion*
*section rather than being presented as a part of rigid results.*

**Reply:** According to this comment, we have modified the manuscript as follows:

(i) Regarding the limitation of the proposed method, we have added an explanation in the Discussion
section as stated above. We have also modified the Abstract as follows:

(original)

“... We found that tissue dynamics are well conserved across species under this space-time coordinate
system, and that the tissue dynamics of both species are mapped with each other through a time-variant
linear transformation in real physical space...”

(revised)

“... We found that tissue dynamics are well conserved across species under this space-time coordinate
system, at least from the early stages of limb development through the phase when basic digit
patterning was established. For this developmental period, we also revealed that the tissue dynamics
of both species are mapped with each other through a time-variant linear transformation in real

physical space,..."

(ii) Secondly, according to the reviewer's suggestion, we have shifted the last section in the Results to
the Discussion.

3) Figure 4B: The legend states that the white points indicate the closest chick stage to each *Xenopus*
stage, and the black points indicate the *Xenopus* to chick stage. However, in Figure 4B, the
representation of white and black points seems to be opposite to the description in the legend. Please
verify the correctness.

**Reply:** Thank you for this comment. We revised the legend as follows:

(Original)

"The white and gray points/curve indicate the closest chick stage to each *Xenopus* stage, which defines
the map $\psi_C \circ \psi_X^{-1}$. The black and gray points/curve show the closest *Xenopus* stage to each chick
stage, defining the map $\psi_X \circ \psi_C^{-1}$."

(Revised)

"The white and gray points/curve indicate the correspondence of each *Xenopus* stage to the closest
chick stage, which defines the map $\psi_C \circ \psi_X^{-1}$. The black and gray points/curve show the
correspondence of each chick stage to the closest *Xenopus* stage, defining the map $\psi_X \circ \psi_C^{-1}$."

4) Figure 4E: The authors mentioned that the choice of initial times for each target affects the level of
similarity, but they only show the case when the *Xenopus* stage is fixed. It would be beneficial to
present the results in a round-robin manner, considering different initial times for each target. Also, it
would be helpful to provide a brief explanation or interpretation as to why the difference level is
smaller at chick stage 21 compared to earlier or later stages. The authors should describe what the
error bars represent and provide the number of samples used in the analysis in the legend.

**Reply:** In this revision, we have analyzed the case when the chick stage is fixed. As shown in the
original manuscript, we found that chick stage 21 was the best when the *Xenopus* stage was fixed at
50.6, so in an additional analysis, we fixed the chick stage at 21 and changed the *Xenopus* stage. As a
result, the best agreement of trajectories between species was found at 50.6 (51.1 was equally good).
Therefore, in this study, we chose 50.6 (*Xenopus*) and 21 (chick) as the initial time combination. Note
that as shown in the heatmap in Fig. 4B, the closest chick stage to *Xenopus* stages 50.6 and 51.1 is 21.
This is thought to be because the relative positions of cells within the tissue do not change very much

between 50.6 and 51.1.

The blue area of the heatmap narrows (like an hourglass) at around chick stage 23-24 and *Xenopus*
stage 52-52.5. In those stages, AP-asymmetric tissue growth clearly appears, and it greatly affects the
change in the rescaled cell position in the ξ space (i.e., cell flow), so those time windows should
correspond between the two species. Therefore, it is necessary to choose an initial time combination
from the stages before those windows. Also, St21 and St50.6 are considered to be a good combination
for the initial time because the PD-AP aspect ratios of the limb bud are similar.

In response to the reviewers' comments, we have revised the statistics somewhat in this revision. The
error bar shows the variation in the distance between species for 24 combinations of α and β values
when the region to be analyzed within the *Xenopus* limb bud is parameterized by α , β , and θ , as shown
in Figure 4H. For α , values corresponding to the position around the boundary between the prospective
zeugopod and stylopod were selected, and for β , values corresponding to the orientation around the A-
P axis were selected. The sizes of the error bars indicate that the interspecific distance is robust for
changes in the values of α and β .

Based on the above, the following sentences have been added to the revised manuscript.

(i) In the Results subsection “Cross-species synchronization of developmental clocks”:

“Further, the blue area of the heatmap narrows (like an hourglass) at around chick stage 23-24 and
*Xenopus* stage 52-52.5. In those stages, A-P asymmetric tissue growth clearly appears, and it greatly
affects the change in the rescaled cell position in the ξ space (i.e., cell flow), meaning that those time
windows should correspond between the two species.”

(ii) In the Materials and Methods section:

“Here, we examined the following two cases: First, the initial time for *Xenopus* was fixed at $t_{Xenopus}$
=50.6, and four initial times, t_{Chick} =20, 21, 23, 24, were tested in the calculations for the chick cell
trajectories in the ξ -space. For each initial value for chick, $\min_{\theta}[\eta(\alpha, \beta, \theta)]$ was calculated for 24
different parameter sets (α, β) , where values corresponding to the position around the boundary
between the prospective zeugopod and stylopod were chosen for α , and values corresponding to the
orientation around the A-P axis were chosen for β . As a result, t_{Chick} =21 was the best fit for the
initial time corresponding to $t_{Xenopus}$ =50.6 (Fig. 4E). In Fig. 4E, the medians and standard deviations
are shown for the 24 sets of (α, β) . The standard deviations are not large, showing the robustness
of the values of $\min_{\theta}[\eta(\alpha, \beta, \theta)]$. Second, we performed a similar analysis for the case in which
the initial time for chick was fixed at t_{Chick} =21. The five initial times, $t_{Xenopus}$ =50.6, 51.1, 51.6, 52.0,
52.4 were tested in the calculations for the *Xenopus* cell trajectories in the ξ -space. For each initial

value for *Xenopus*, $\min_{\theta}[\tilde{\eta}(\alpha, \beta, \theta)]$ was calculated for the same 24 parameter sets for (α, β)
as above, where $\tilde{\eta}$ is defined as follows:

$$700 \quad \tilde{\eta}(\alpha, \beta, \theta) = \left\langle \delta(t_{\text{Chick}}^k, t_{\text{Xenopus}}^{k, \text{CtoX}}(\alpha, \beta, \theta)) \right\rangle_k,$$

$$701 \quad t_{\text{Chick}}^k \in [21, 30.5].$$

Consequently, the best agreement of trajectories between the two species was found at $t_{\text{Xenopus}} = 50.6$
(51.1 was equally good). Based on these results, we chose $t_{\text{Xenopus}} = 50.6$ and $t_{\text{Chick}} = 21$ as the initial
time combination. Note that the aspect ratios (P-D vs A-P) of the limb buds (for the prospective
autopod and zeugopod) of both species are similar at $t_{\text{Xenopus}} = 50.6$ and $t_{\text{Chick}} = 21$.”

5) *Figure 5: The authors claim a high degree of similarity in ζ -space (line 353), which is an important*
*result. It would be beneficial to quantify the level of similarity to provide a more quantitative*
*assessment of the observed similarity.*

**Reply:** After careful consideration, we have decided to describe only qualitative similarities here. As
mentioned in the Discussion section, recent advances in spatial transcriptome techniques are
remarkable. In fact, we have recently started a spatial transcriptome analysis of chick and *Xenopus*
limb development for the purpose of evaluating the similarity of expression patterns in the τ - ζ
coordinate system between the two species in a quantitative and genome-wide manner. Clear data on
this point will be obtained in the future.

6) *Materials and Methods, line 441: “All animal handling was performed under appropriate*
*anesthesia ..”. It is recommended to provide a detailed description of the anesthesia protocol used in*
*the study for the purpose of reproducibility.*

**Reply:** Thank you for your comment. We have added accurate chemical information about anesthesia
as follows:

“0.05% tricaine methanesulfonate dissolved in Holtfreter’s solution (60 mM NaCl, 0.6 mM KCl, 0.9
mM CaCl₂, 2.4 mM NaHCO₃)”.

REVIEWERS' COMMENTS

Reviewer #1 (Remarks to the Author):

The authors have provided a helpful response to both the major and minor points raised in the first round of review, and amended the manuscript accordingly. I am happy to recommend publication of this revised version.

Reviewer #2 (Remarks to the Author):

Reviewing the comments from the revision of this manuscript, the authors have attended to all of them fairly carefully so I would support publication. The only thing I would strongly suggest is recognizing that the word 'conservation' as they use it implies that the trait of interest is not changed from the shared ancestor (i.e. they say the limb development process is conserved between frog and chick), while in this case, I believe evolutionary convergence is at least equally likely. It doesn't lessen the results that similar processes are involved and that those processes are present because of deeply conserved genetic, cellular and tissue properties, but the way the frog limb evolved coincident with a novel life history with metamorphosis, is unknown, therefore convergence should be mentioned in the discussion as an alternative explanation in addition to simply perpetuated conservation of limb developmental processes. I think the frog development literature supports this (the refs I suggested earlier). This perspective might also slightly change the discussion in other places, something for the authors to consider for consistency through the manuscript.

Second - a comment on line 503-504. I suggest adding "'species-specific' features of organ morphology' to clarify.

Reviewer #3 (Remarks to the Author):

The authors clarified all of my questions and concerns. I believe that the manuscript is now much improved and ready for publication. The revisions made have significantly enhanced the clarity and overall quality of the work.

Reply to the reviewer 2's comments

Reviewing the comments from the revision of this manuscript, the authors have attended to all of them fairly carefully so I would support publication. The only thing I would strongly suggest is recognizing that the word 'conservation' as they use it implies that the trait of interest is not changed from the shared ancestor (i.e. they say the limb development process is conserved between frog and chick), while in this case, I believe evolutionary convergence is at least equally likely. It doesn't lessen the results that similar processes are involved and that those processes are present because of deeply conserved genetic, cellular and tissue properties, but the way the frog limb evolved coincident with a novel life history with metamorphosis, is unknown, therefore convergence should be mentioned in the discussion as an alternative explanation in addition to simply perpetuated conservation of limb developmental processes. I think the frog development literature supports this (the refs I suggested earlier). This perspective might also slightly change the discussion in other places, something for the authors to consider for consistency through the manuscript.

Reply to the comment (1)

Thank you very much for this advice. We have added the following sentences into the Discussion section of the second revised manuscript:

“It should be noted that, in the context of evolution, it is important to exercise caution when using the term 'conservation.' Generally, the high similarities in tissue dynamics and associated genes make it plausible that these processes have been conserved without change from a common ancestor. However, it is essential to remain open to the possibility that they may have independently evolved through convergence.”

Second - a comment on line 503-504. I suggest adding "'species-specific' features of organ morphology' to clarify.

Reply to the comment (2)

We have added the word in the Discussion section.